# The methyl phosphate capping enzyme Bmc1/Bin3 is a stable component of the fission yeast telomerase holoenzyme

Jennifer Porat[1], Moaine El Baidouri [2,3], Jorg Grigull[4], Jean-Marc Deragon [2,3,5] & Mark A. Bayfield [1✉]

The telomerase holoenzyme is critical for maintaining eukaryotic genome integrity. In addition to a reverse transcriptase and an RNA template, telomerase contains additional proteins that protect the telomerase RNA and promote holoenzyme assembly. Here we report that the methyl phosphate capping enzyme (MePCE) Bmc1/Bin3 is a stable component of the *S. pombe* telomerase holoenzyme. Bmc1 associates with the telomerase holoenzyme and U6 snRNA through an interaction with the recently described LARP7 family member Pof8, and we demonstrate that these two factors are evolutionarily linked in fungi. Our data suggest that the association of Bmc1 with telomerase is independent of its methyltransferase activity, but rather that Bmc1 functions in telomerase holoenzyme assembly by promoting TER1 accumulation and Pof8 recruitment to TER1. Taken together, this work yields new insight into the composition, assembly, and regulation of the telomerase holoenzyme in fission yeast as well as the breadth of its evolutionary conservation.

---

[1] Department of Biology, York University, Toronto, Canada. [2] LGDP-UMR5096, Université de Perpignan Via Domitia, Perpignan, France. [3] CNRS LGDP-UMR5096, Perpignan, France. [4] Department of Mathematics and Statistics, York University, Toronto, Canada. [5] Institut Universitaire de France, Paris, France. ✉email: bayfield@yorku.ca

To ensure complete DNA replication, the termini of eukaryotic chromosomes contain tandem repeats, or telomeres, that can be continually extended as the ends of linear chromosomes are lost through DNA replication[1]. Telomeres serve as protection from DNA degradation, end-to-end fusions, chromosomal rearrangements, and chromosome loss[1,2]. With very few exceptions, eukaryotes extend telomeric DNA sequences through the telomerase holoenzyme, a complex containing the telomerase reverse transcriptase (Trt1 in the fission yeast Schizosaccharomyces pombe), an RNA template (TER1 in S. pombe), and accessory proteins that promote complex assembly and tethering to telomeric DNA[1–5]. While the reverse transcriptase/RNA template core of telomerase is generally well-conserved among eukaryotes, major differences exist, including in the sequence and structure of the RNA template. In contrast to ciliate telomerase RNAs, which are small RNA polymerase III transcripts[6], yeast and metazoan telomerase RNAs are longer and transcribed by RNA polymerase II as precursors[7,8]. These precursors then undergo an extensive maturation process to yield the mature form that integrates into the telomerase holoenzyme[3,4,7,8].

In S. pombe, the mature form of TER1 is generated from an intron-containing precursor through a spliceosome-catalyzed reaction involving the release of the 5' exon prior to exon ligation[9]. TER1 maturation then proceeds with the sequential binding of the Sm and Lsm complexes. Sm proteins, well-characterized for their role in splicing, bind TER1 immediately upstream the 5' splice site and promote 3' maturation and the addition of a 5' trimethylguanosine (TMG) cap by the methyltransferase Tgs1[10]. The Sm complex is then replaced by the Lsm2-8 complex, which serves to protect the mature 3' end of TER1 from exonucleolytic degradation and promotes the interaction between TER1 and Trt1 in the active telomerase holoenzyme. The switch from the Sm to Lsm complexes correlates with distinct, nonoverlapping substrates: TER1 precursors are exclusively bound by the Sm complex, while the mature form of TER1, ending in a polyuridylate stretch upstream the spliceosomal cleavage site, is only bound by the Lsm2-8 complex[10].

More recently, a role has been proposed for Pof8, a La-related protein 7 (LARP7) homolog, in telomerase assembly and telomere maintenance in S. pombe. Ciliate LARP7-family proteins, including p65 from Tetrahymena thermophila, have previously been characterized for their functions in telomerase assembly[11]. Binding of p65 to stem IV of the T. thermophila telomerase RNA TER results in a conformational change to the RNA and subsequent binding of the reverse transcriptase TERT, suggesting a p65-dependent hierarchical assembly of the telomerase holoenzyme[12]. Pof8 is hypothesized to function similarly in S. pombe, with Pof8 binding to TER1 promoting recruitment of the Lsm2-8 complex and enhancing the interaction between TER1 and Trt1. Accordingly, Pof8 deletion strains possess shorter telomeres indicative of a defect in telomerase function[13–15]. Pof8 also functions in telomerase RNA quality control through recognition of the correctly folded pseudoknot region and subsequent recruitment of the Lsm2-8 complex to protect the 3' end from degradation[16]. Telomerase-related functions in LARP7-family proteins, particularly Pof8, map to a conserved C-terminal domain characteristic of LARP7-family members, the extended RNA-recognition motif (xRRM)[17,18]. The Pof8 xRRM is a major determinant for Pof8-mediated telomerase activity, contributing to TER1 binding and complex assembly[14,19]. As other fungal Pof8 homologs also possess an xRRM, it is anticipated that Pof8 may have a conserved role in telomere maintenance beyond fission yeast[14,20].

Conversely, LARP7 homologs in higher eukaryotes are best characterized in the transcriptional regulatory 7SK snRNP, which includes the RNA polymerase III-transcribed 7SK snRNA, the methyl phosphate capping enzyme (MePCE), and hexamethylene bisacetamide-inducible protein (HEXIM1/2). The 7SK snRNP inhibits transcription by sequestering positive transcription elongation factor (P-TEFb) and preventing it from phosphorylating the C-terminal domain of the largest subunit of RNA polymerase II, which is associated with the transition into productive transcription elongation[21–23]. The La module and xRRM of LARP7 bind 7SK through its polyuridylate trailer and the 3' hairpin, respectively, and promote MePCE recruitment to the complex[24,25]. In addition to adding a γ-monomethyl phosphate cap (i.e., CH$_3$-pppN) to 7SK snRNA as a means of protecting it from 5' exonucleolytic degradation, MePCE remains stably bound to 7SK snRNA to stabilize the complex[24]. MePCE, a homolog of the Drosophila melanogaster Bin3/BCDIN3 protein, also catalyzes the addition of this atypical cap structure to U6 snRNA[26]. Bin3/MePCE homologs are present in many eukaryotes, but to date little has been studied outside of humans and Drosophila[27]. More recent work has identified BCIND3D, a related Bin3-family protein overexpressed in breast cancer cells, as the enzyme responsible for methylating the 5' monophosphate of histidine transfer RNA and pre-miR-145[28,29]. Despite the many insights into the role of Bin3/MePCE in RNA processing in higher eukaryotes, the function of the S. pombe Bin3 homolog has yet to be determined.

In this work, we have explored the RNA targets of the S. pombe Bin3 homolog (henceforth referred to as Bmc1; Bin3/MePCE1) in an effort to uncover its function. In addition to an expected association of Bmc1 with U6 snRNA, we present data showing that Bmc1 is a stable component of the S. pombe telomerase RNP. Our results also provide evidence for an evolutionarily conserved interaction between Bmc1 and LARP7 homologs. Importantly, we show that the Bmc1–Pof8 interaction is required for the recruitment of Bmc1 to both U6 snRNA and the active telomerase holoenzyme. Additionally, we provide data that Bmc1 does not catalyze the addition of a γ-monomethyl phosphate cap on TER1, suggesting a catalytic independent function. Rather, we present evidence that Bmc1 functions in concert with Pof8 to promote telomerase assembly and TER1 accumulation. Taken together, our results identify Bmc1 as a component of the S. pombe telomerase holoenzyme, thus adding a new layer of complexity to telomerase assembly and telomere maintenance.

## Results

**S. pombe Bmc1 interacts with U6 snRNA and the telomerase RNA TER1.** While Bmc1/MePCE is conserved in many eukaryotes, its presence in the fission yeast S. pombe, an organism lacking 7SK snRNA[27,30], is not well understood. Unlike the longer Bmc1/MePCE homologs from higher eukaryotes, S. pombe Bmc1 only contains a conserved methyltransferase/SAM-binding domain (Supplementary Fig. 1A). Alignment of the methyltransferase domain of S. pombe Bmc1 to that of H. sapiens MePCE and D. melanogaster Bin3 reveals that previously identified residues critical for SAM and SAH binding and nucleotide binding[31] are highly conserved in S. pombe (Supplementary Fig. 1B, C).

We first sought to determine the RNA substrates of S. pombe Bmc1 to better understand its role in the processing of fission yeast noncoding RNA(s). Following the integration of a protein A (PrA) tag into the Bmc1 genomic locus, we performed RNA immunoprecipitation coupled to sequencing (RIP-Seq), followed by validation of potential candidate substrates with northern blotting and semi-quantitative RT-PCR (Fig. 1 and Supplementary Data 1). Consistent with previous reports of MePCE interacting with U6 snRNA[26], U6 emerged as one of the most highly enriched Bmc1-interacting RNA transcripts in our Seq dataset (Fig. 1A and Supplementary Data 1). S.

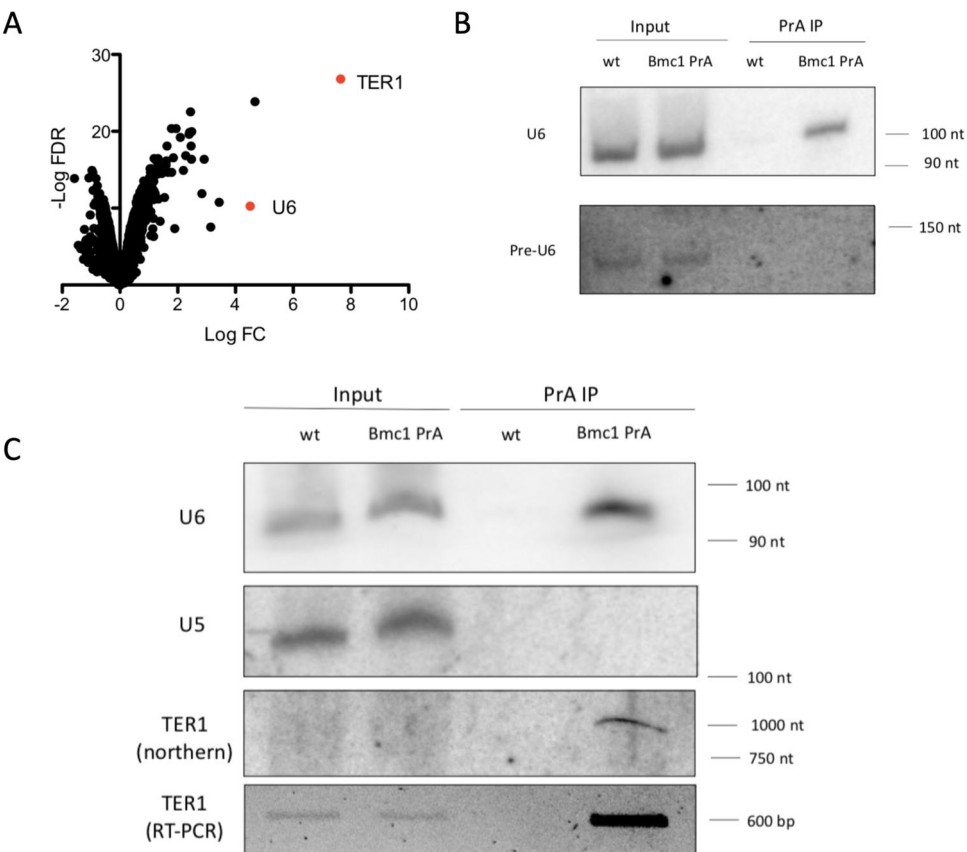

**Fig. 1 Bmc1 interacts with U6 snRNA and the telomerase RNA TER1. A** Enrichment of Bmc1 PrA immunoprecipitated transcripts compared to an untagged control (biological replicates = 3). Axes represent log2 of fold change (FC) and negative log of false discovery rate (FDR) (Benjamini–Hochberg adjusted *P* value ≤0.05). **B** Northern blot analysis of the mature and intron-containing U6 from total RNA and PrA immunoprecipitates from untagged (wild type, wt) and PrA-tagged Bmc1 strains. **C** Northern blot of the telomerase RNA TER1, U6, and U5 from total RNA and PrA immunoprecipitates from untagged and PrA-tagged Bmc1 strains, and semi-quantitative RT-PCR analysis of TER1. Source data are provided as a Source Data file.

*pombe* U6 is an unusual transcript in that it is transcribed by RNA polymerase III yet contains an mRNA-type intron removed by the spliceosome[32,33]. Alignment of RIP-Seq reads to the *S. pombe* genome suggested that Bmc1 interacts exclusively with the spliced form of U6. We confirmed this with northern blotting, demonstrating robust immunoprecipitation of mature U6 and no immunoprecipitation of the intron-containing precursor (Fig. 1B). Chromatin immunoprecipitation studies suggest that 7SK and U6 snRNA may be co-transcriptionally modified by the human MePCE homolog[34], however, our data are consistent with a model wherein *S. pombe* Bmc1 interacts with U6 post-splicing.

Surprisingly, the *S. pombe* telomerase RNA TER1 was the most highly enriched hit in the Bmc1 immunoprecipitates (Fig. 1A, C). Given the previously established link between the human Bmc1 homolog MePCE and human LARP7[23], as well as the established link between the *S. pombe* LARP7 homolog Pof8 and the telomerase holoenzyme[13–15], we considered the possibility that Bmc1 may also be part of the telomerase holoenzyme through an evolutionarily conserved interaction with LARP7-family members.

**Bmc1 interacts with the mature form of TER1.** Since TER1 processing and maturation involves a spliceosome-catalyzed reaction[9], we next set out to determine the processing state of Bmc1-associated TER1 by sequencing the 5' and 3' ends through circularized rapid amplification of cDNA ends (cRACE) (Fig. 2A). We also performed cRACE on TER1 immunoprecipitated by Pof8, the telomerase reverse transcriptase Trt1, and Lsm3, all known

components of the mature TER1-containing active telomerase holoenzyme[3,4,10,13–15]. All sequenced candidates had a discrete 5' end consistent with the reported 5' end[3,4] and a similar distribution of 3' ends ending immediately upstream the 5' splice site[9], suggesting that Bmc1 interacts with the full-length, 3' end-matured TER1 associated with the active telomerase holoenzyme. The heterogeneity observed at the 3' end of all candidates is likely the result of exonucleolytic nibbling prior to binding of the Lsm2-8 complex[10]. To investigate this further, we subjected TER1 RNA immunoprecipitated by Bmc1, Trt1, Pof8, and Lsm3 to RNase H cleavage to generate shorter 5' and 3' fragments and compared fragment size by northern blot (Fig. 2B). The sizes of the 5' and 3' ends of TER1 were similar across immunoprecipitates, further substantiating that Bmc1 binds the mature form of TER1, rather than the intron-containing precursor.

Further TER1 processing in *S. pombe* involves the addition of a 5' trimethylguanosine (TMG) cap by Tgs1[10]. Since Bmc1 homologs are normally associated with the formation of a γ-monomethyl phosphate cap, and to further understand when Bmc1 interacts with TER1 with respect to TER1 processing, we examined the cap structure of Bmc1-associated TER1 by immunoprecipitating RNA associated with Bmc1 and subsequently re-immunoprecipitating this RNA with an anti-TMG antibody. The specificity of the anti-TMG antibody was confirmed by showing that it effectively enriched the TMG cap-containing U5 snRNA over the TMG cap-lacking U6 snRNA or an unrelated tRNA (Fig. 2C). While Bmc1-associated U6 snRNA was not immunoprecipitated by the anti-TMG antibody, in

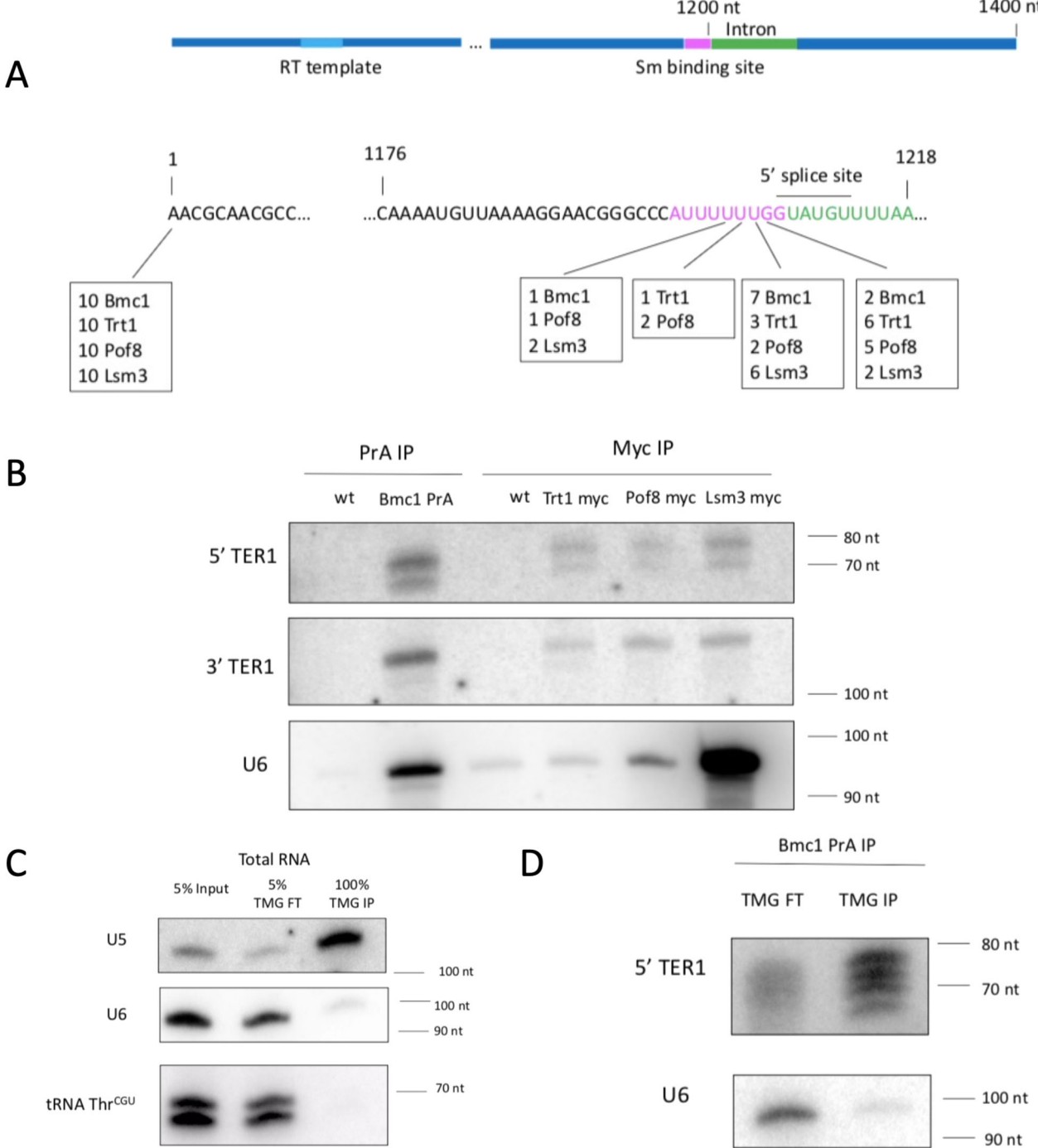

**Fig. 2 Bmc1 interacts with the same TER1 species as well-established components of the telomerase holoenzyme. A** The 5′ and 3′ ends of Bmc1-, Trt1-, Pof8-, and Lsm3-associated TER1 were identified by cRACE. The results of ten independent clones per immunoprecipitation are shown below a schematic of the architecture of TER1. **B** RNase H northern blots of RNase H-generated 5′ and 3′ ends of TER1 immunoprecipitated by various telomerase components. The same blot was stripped and reprobed for U6. **C, D** Northern blot of α-TMG flow through (FT) and immunoprecipitated (IP) transcripts from total RNA (**C**) and Bmc1-associated RNA (**D**). Source data are provided as a Source Data file.

agreement with data demonstrating the presence of a γ-monomethyl phosphate cap on U6 in *S. pombe*[35], Bmc1-associated TER1 RNA was effectively enriched by anti-TMG immunoprecipitation (Fig. 2D). These data are consistent with previous work demonstrating a TMG cap on TER1 in *S. pombe*, and also with Bmc1 associating with TER1 following spliceosomal cleavage and 5′ TMG capping, similar to what has been posited for Pof8[13]. This suggests that Bmc1 interacts with the primary cohort of TMG-capped TER1 transcripts, and that TER1

is not a substrate for Bmc1-catalyzed γ-monomethyl phosphate capping.

**Bmc1 interacts with components of the mature telomerase holoenzyme.** Since our results indicate that Bmc1 interacts with the mature form of TER1, we set out to confirm the presence of Bmc1 in the *S. pombe* telomerase holoenzyme. We immunoprecipitated Bmc1 and identified interacting proteins through mass spectrometry, followed by validation using co-immunoprecipitation (Fig. 3).

We identified Pof8, the Lsm2-8 complex, and Trt1 in Bmc1 immunoprecipitates, all of which make up the catalytically active core telomerase holoenzyme[5,10,13–15,36,37] (Supplementary Data 2). We also detected the RNase P and RNase MRP subunit Pop100 in Bmc1 immunoprecipitations. Pop1, Pop6, and Pop7, subunits of RNase P and RNase MRP in budding yeast, have recently been identified as components of the budding yeast telomerase complex[38–40], suggesting that the involvement of RNase P and RNase MRP subunits in telomere maintenance is evolutionarily conserved. The finding that Bmc1 interacts with core components of the telomerase holoenzyme, and not the TMG capping enzyme Tgs1, suggests Bmc1 does indeed associate with the active telomerase holoenzyme rather than a precursor. The involvement of Bmc1 in the telomerase holoenzyme was further substantiated by gene ontology analysis to determine overrepresented biological processes and cellular components among the top 50 Bmc1 interactors. Telomerase holoenzyme complex assembly and telomere maintenance via telomerase emerged as the top overrepresented biological processes (Fig. 3A). Similarly, overrepresented cellular components include the Lsm2-8 complex, the telomerase holoenzyme, and spliceosomal snRNPs (Supplementary Table 1). We also repeated immunoprecipitations in the presence of benzonase to determine if these interactions are direct or mediated through a nucleic acid intermediate. With the exception of Pof8, certain members of the Lsm2-8 complex, and an uncharacterized protein (SPCC18B5.09c, recently identified as the telomerase component Thc1[41]), interactions between Bmc1 and other components of the telomerase holoenzyme were lost, suggestive of an interaction mediated by TER1 (Supplementary Data 2).

We validated the interactions between Bmc1 and Pof8, as well as Bmc1 and Trt1, through co-immunoprecipitation (Fig. 3B, C and Supplementary Fig. 2). We also performed co-immunoprecipitation with benzonase, demonstrating that the interaction between Bmc1 and Pof8 persists with benzonase treatment, while the Bmc1–Trt1 interaction is lost (Fig. 3B, C). To confirm that the Bmc1–Trt1 interaction is mediated by TER1, we repeated co-immunoprecipitation in a TER1 knockout strain. The direct interaction between Bmc1 and Pof8 remained intact in the absence of TER1, while the Bmc1–Trt1 interaction was completely lost, indicative of Bmc1 assembly in the telomerase holoenzyme nucleated, in large part, by the telomerase RNA itself. The RNA-dependence of the interaction between Bmc1 and Trt1 is reminiscent both of the budding yeast telomerase RNA TLC1, which acts as a scaffold for telomerase holoenzyme assembly[42], as well as the MePCE/Bin3-containing 7SK snRNP[43]. The finding that Bmc1 and Pof8 interact directly is in agreement with previous reports demonstrating a protein–protein interaction between MePCE and LARP7 in the context of the vertebrate 7SK snRNP[34,43], suggesting the direct interaction may be evolutionarily conserved.

**Bmc1 is recruited to the active telomerase holoenzyme through the LARP7-family protein Pof8.** We then tested whether Bmc1 was recruited to the telomerase holoenzyme through its interaction with Pof8, much like the human LARP7 homolog promotes MePCE recruitment to 7SK snRNA[24]. Immunoprecipitation of Bmc1 in the context of a pof8Δ strain resulted in a complete loss of TER1 binding, providing evidence that the interaction between Bmc1 and TER1 is dependent on an interaction between Bmc1 and Pof8 (Fig. 4A, B). Unexpectedly, the interaction between Bmc1 and U6 was also lost in the pof8Δ strain. Recent findings indicate that mammalian LARP7 binds U6 to guide post-transcriptional modification[44,45], so it is tempting to speculate that Pof8 may also mediate Bmc1 binding to U6 in fission yeast. To address this, we immunoprecipitated Pof8 alongside Bmc1 and other components of the telomerase holoenzyme and looked for U6 enrichment. We observed slight enrichment of U6 in Pof8-myc immunoprecipitates relative to immunoprecipitation using an untagged strain or the Trt1-myc-tagged strain (Fig. 4C). We did not see enrichment for the intron-containing precursor, suggesting that like Bmc1, Pof8 interacts with the spliced form of U6. The slight enrichment of U6 with Pof8 immunoprecipitation, compared to the more robust immunoprecipitation seen with Bmc1 and Lsm3, suggests that Pof8 may transiently bind U6, perhaps serving to load Bmc1 on U6. Since LARP7 binding has been reported to disrupt the catalytic activity of MePCE[34], a transient Pof8-U6 interaction may be consistent with U6 receiving a γ-monomethyl phosphate cap, compared to TER1, where stable Pof8 binding may prevent Bmc1 catalytic activity.

To further investigate how the presence of Bmc1 in the TER1-containing telomerase holoenzyme is reliant on Pof8, we fractionated lysates from a wild type and a pof8Δ strain on a glycerol gradient and analyzed the sedimentation patterns of Bmc1, Pof8, and TER1 (Supplementary Fig. 3). We found that Bmc1 and Pof8 co-sedimented with TER1 in higher molecular weight fractions in a wild-type strain. In contrast, Bmc1 was depleted from higher molecular weight fractions containing TER1 in the absence of Pof8. We also noted a slight shift in TER1 toward lighter fractions in the Pof8 deletion strain, which could be due to a lighter-migrating telomerase holoenzyme lacking Bmc1 and Pof8. These data are also consistent with Bmc1 stably associating with the telomerase holoenzyme in a manner that is dependent on Pof8.

Knowing that Bmc1 interacts with TER1 and components of the telomerase holoenzyme, we next set out to confirm whether Bmc1 is part of the catalytically active telomerase holoenzyme. We performed a previously described in vitro telomerase assay that relies on the presence of the TER1- and Trt1-containing telomerase holoenzyme to extend an oligonucleotide resembling telomeric DNA[13] (Fig. 4D). Bmc1 immunoprecipitates extended the oligonucleotide in a similar manner previously demonstrated for Pof8 (see ref. [13] and Fig. 5D), as well as showed the same loss of activity upon RNase A treatment, supporting the idea that Bmc1, much like Pof8, is a component of the active telomerase holoenzyme (Fig. 4D). Consistent with previous results[13,16], we also observed a loss of activity for Bmc1 immunoprecipitated from a pof8Δ strain, which can largely be attributed to the loss of functional, correctly assembled telomerase occurring in the absence of Pof8. Since Trt1 is only recruited to the telomerase holoenzyme following Pof8 and Lsm2-8 binding, coupled with the idea that the assay relies on the presence of reverse transcriptase, these results are consistent with a model in which Bmc1 is recruited to TER1 through its interaction with Pof8 and remains bound through subsequent holoenzyme assembly and the catalytic cycle (Fig. 4E).

To further investigate the link between Bmc1 and Pof8 in telomerase holoenzyme assembly, we tested whether overexpression of Bmc1 could rescue the short telomere phenotype of a pof8Δ strain[13–15]. Consistent with a model in which Bmc1 requires Pof8 for its assembly into the active telomerase holoenzyme, Bmc1 overexpression in a pof8Δ strain was insufficient to rescue telomere shortening, as measured by Southern blot, as well chromosome fusions occurring after telomeres reach a critically short length previously associated with the pof8Δ strain[13] (Supplementary Fig. 4).

**Bmc1 promotes TER1 accumulation and Pof8 recruitment to telomerase.** We then interrogated the functional role of Bmc1 in telomerase by creating a bmc1 knockout strain where we replaced the bmc1 open reading frame with a phleomycin resistance

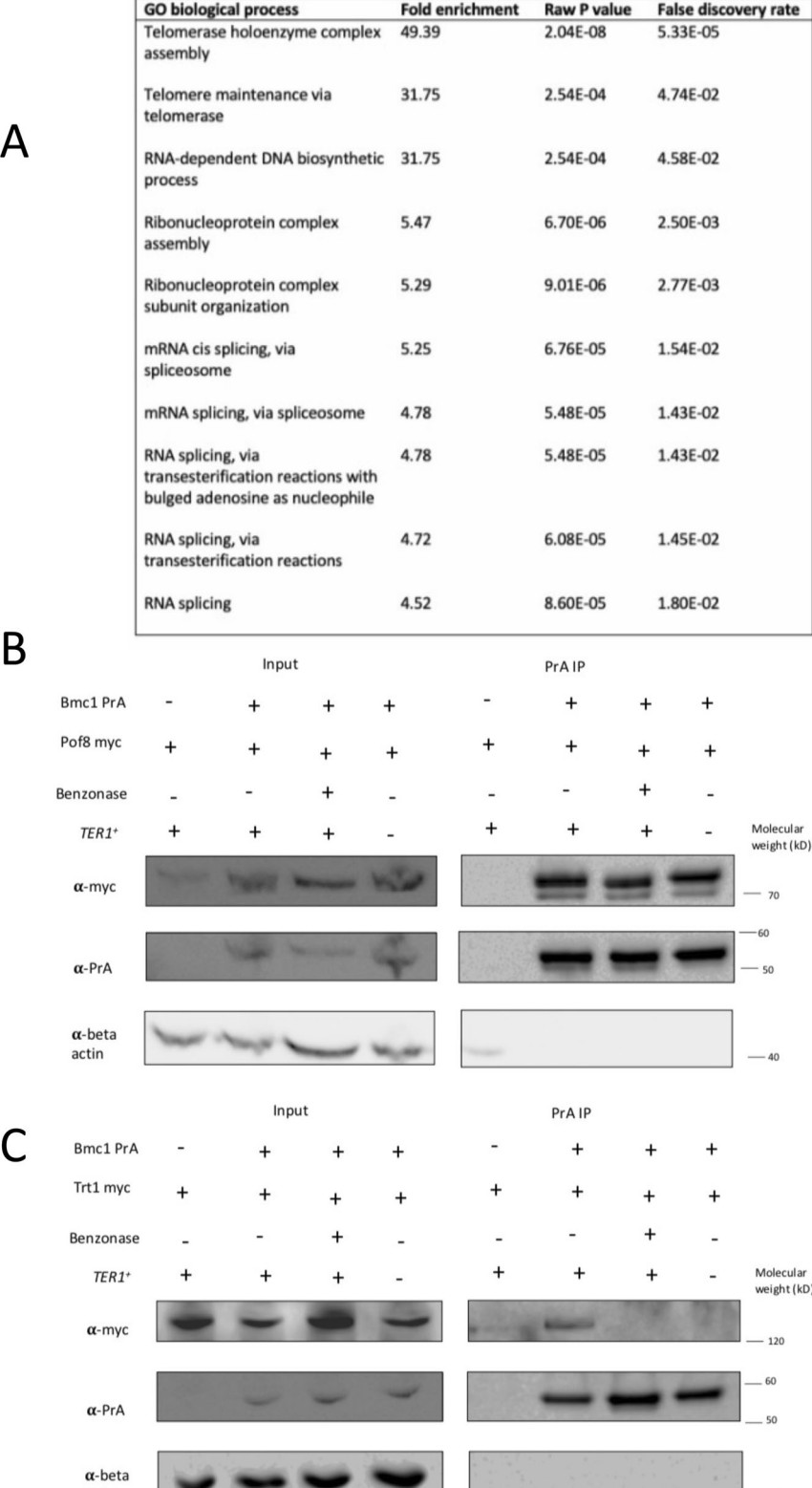

**Fig. 3 Bmc1 interacts with components of the mature telomerase holoenzyme. A** Gene ontology analysis (biological process) of top 50 Bmc1 protein interactors. **B**, **C** Examination and nucleic acid-dependence of interactions between Bmc1 and Pof8 (**B**) and Bmc1 and Trt1 (**C**) by co-immunoprecipitation and western blotting. Blots were reprobed for beta-actin as a loading control. Source data are provided as a Source Data file.

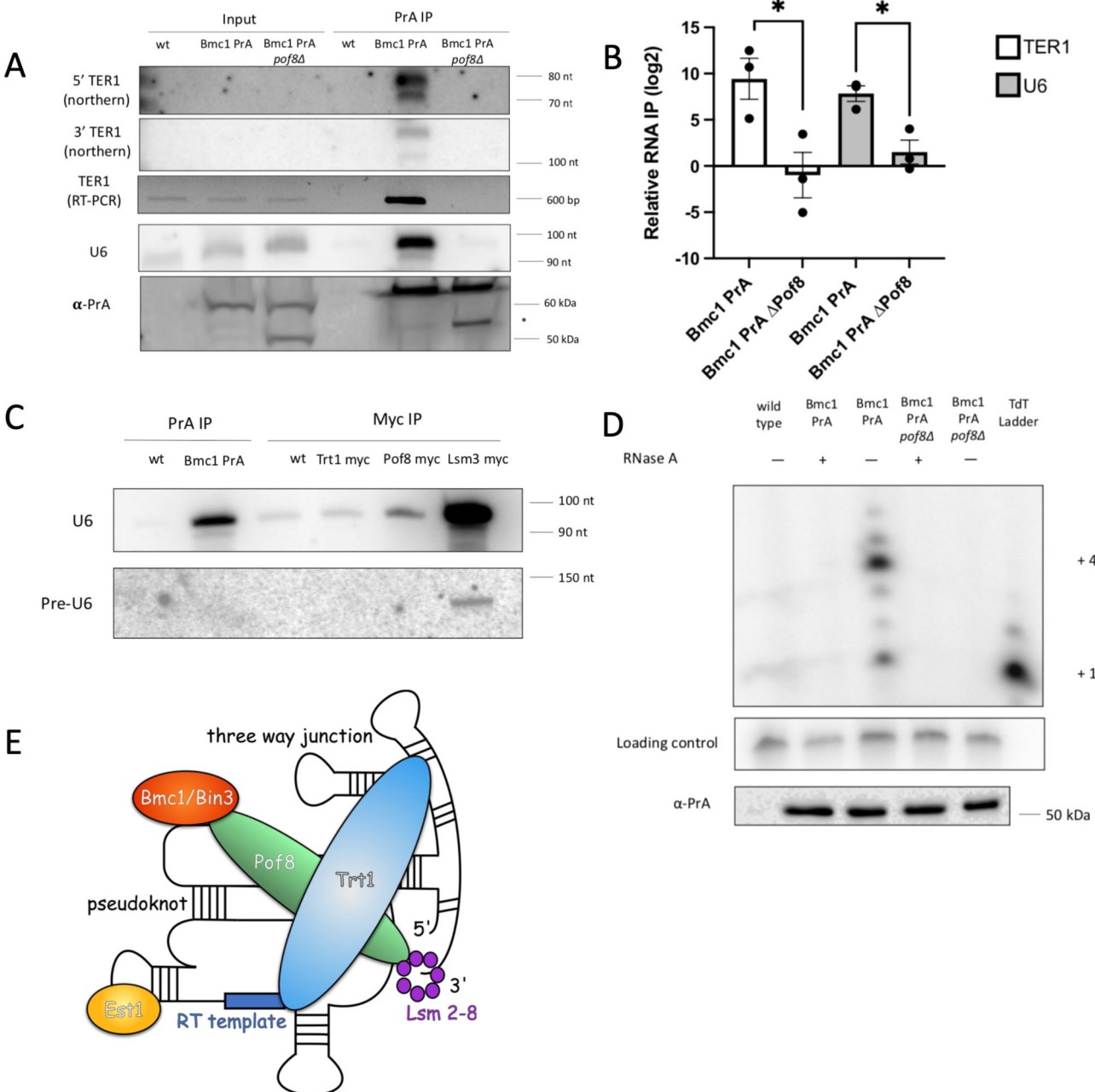

**Fig. 4 Bmc1 is recruited to the active telomerase holoenzyme by Pof8. A** Northern blot and semi-quantitative RT-PCR of TER1 and U6 in PrA immunoprecipitates for an untagged wild type (wt), PrA-tagged (Bmc1 PrA), and PrA-tagged Pof8 knockout strain (Bmc1 PrA ΔPof8). Bmc1 PrA was detected in input and immunoprecipitated samples by western blots probing for PrA (bottom panel). Possible cleavage products are indicated with an asterisk. **B** qRT-PCR of TER1 and U6 in Bmc1 PrA immunoprecipitates from wild-type and *pof8Δ* strains normalized to input RNA. Relative TER1 and U6 IP was calculated by comparing percent immunoprecipitation of TER1 or U6 to immunoprecipitation from an untagged strain (mean ± standard error, two-tailed unpaired *t* test *$P < 0.05$, **$P < 0.01$, ***$P < 0.001$, and ****$P < 0.0001$) ($n = 3$ biological replicates). **C** Northern blot of mature and intron-containing U6 in PrA- and myc-tagged immunoprecipitated RNA. **D** Telomerase assay of PrA-tagged Bmc1 in a wild-type and *pof8Δ* strain. A [32]P-labeled 100-mer oligonucleotide was used as a loading control. Telomerase extension products were compared to a terminal transferase ladder, with +1 and +4 extension products indicated. Western blot probing for PrA following PrA immunoprecipitation is shown in the panel below. **E** Proposed model of the fission yeast telomerase holoenzyme. TER1 structure and binding locations are based on models constructed by Hu et al.[16] and Mennie et al.[14] Source data are provided as a Source Data file.

cassette which was confirmed by PCR and sequencing, as well as qRT-PCR to confirm a lack of *bmc1* mRNA (Supplementary Fig. 5A, B). To address conflicting reports in the literature as to the nature of *bmc1*'s essentiality in *S. pombe*[46,47], we back-crossed our *bmc1* deletion strain with a wild-type strain. Sporulation and tetrad dissections yielded viable haploid colonies possessing phleomycin resistance (Supplementary Fig. 5C), confirming that

*bmc1* is not essential and enabling us to conduct further mechanistic studies.

Consistent with its strong dependence on Pof8 for recruitment to telomerase, *bmc1* deletion appears to decrease steady-state TER1 levels, much like a *pof8* deletion strain, with no reduction in U6 abundance (Fig. 5A). Importantly, re-introduction of plasmid expressed Bmc1 to *bmc1Δ* cells was sufficient to restore wild-type

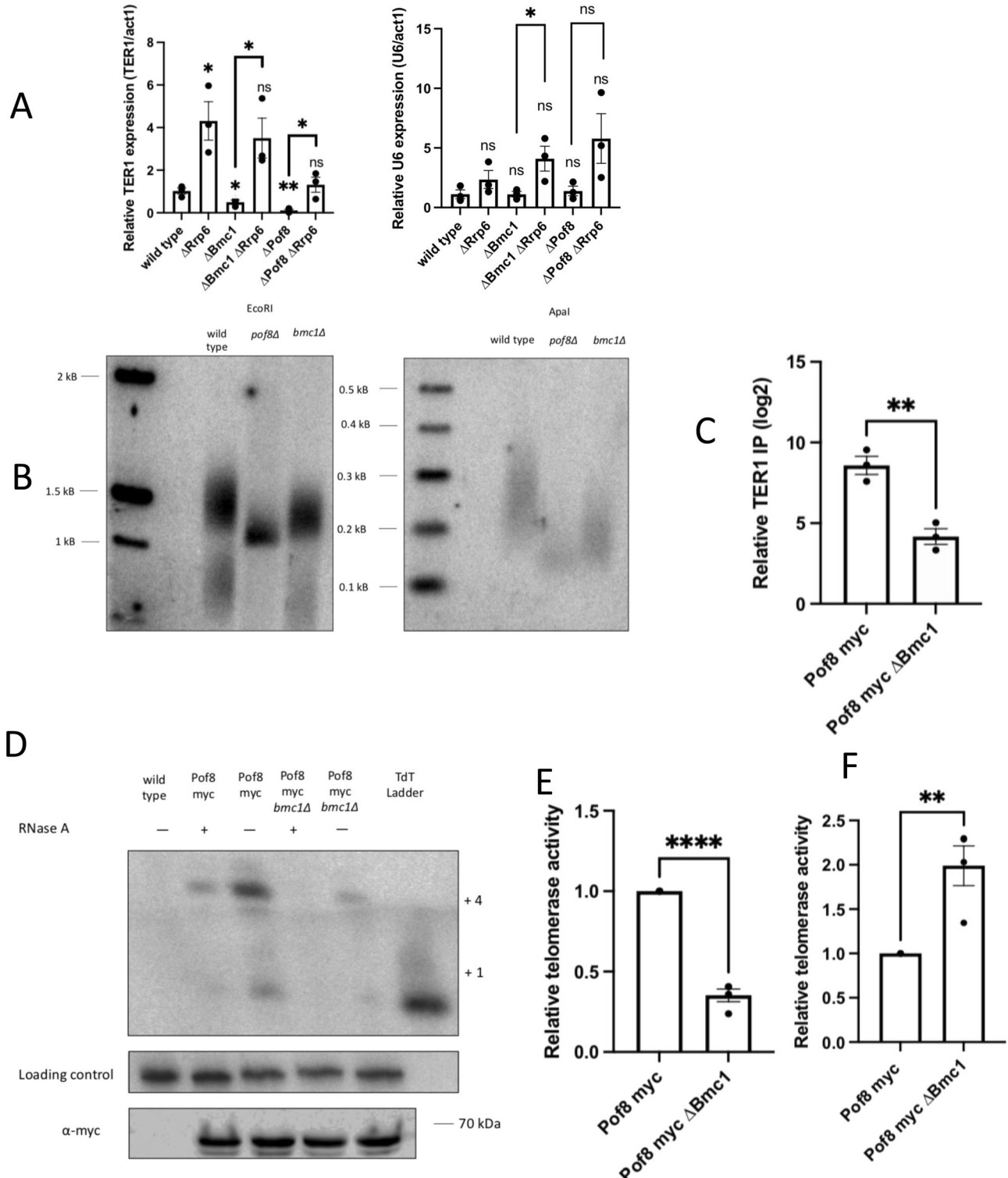

levels of TER1 (Supplementary Fig. 5D). To understand the mechanism by which Bmc1 appears to promote TER1 accumulation, we combined our *bmc1Δ* strain with deletion of *rrp6*, an exonuclease component of the nuclear exosome that has been previously implicated in TER1 degradation[13,15,16,48,49]. We observed a restoration of TER1 levels in the *bmc1Δ rrp6Δ* strain (Fig. 5A), similar to what has been observed for *pof8Δ*, where *rrp6* deletion rescues decreased TER1 levels[15]. From this, we hypothesize that Bmc1 promotes TER1 accumulation by preventing 3' decay by the exosome to maintain steady-state TER1 levels.

In line with decreased TER1 levels, *bmc1* deletion resulted in shorter telomere length compared to a wild-type strain, which became more evident when genomic DNA was digested with ApaI to yield shorter telomere fragments (Fig. 5B). We continued passaging strains to get to the point of crisis (>100 generations), where telomeres are lost and cells either die or circularize their chromosomes, leading to a loss of an observable telomere signal

**Fig. 5 Bmc1 promotes TER1 accumulation and Pof8 recruitment to telomerase. A** Quantitation of TER1 and U6 in total RNA by qRT-PCR, normalized to *act1* mRNA. *P* values over bars represent comparison to a wild-type strain (mean ± standard error, two-tailed unpaired *t* test *$P < 0.05$, **$P < 0.01$, ***$P < 0.001$, and ****$P < 0.0001$) ($n = 3$ biological replicates). **B** Southern blot comparing telomere length in following three restreaks on rich media (one restreak = 20–25 generations). Genomic DNA was digested with EcoRI (left) or ApaI (right) to yield different sized telomere restriction fragments. **C** qRT-PCR of TER1 in Pof8-myc immunoprecipitates from a wild-type and *bmc1Δ* strain normalized to input RNA. Relative TER1 IP was calculated by comparing percent immunoprecipitation of TER1 to immunoprecipitation from an untagged strain (mean ± standard error, two-tailed unpaired *t* test *$P < 0.05$, **$P < 0.01$, ***$P < 0.001$, and ****$P < 0.0001$) ($n = 3$ biological replicates). *P* values over each bar represent results of a two-tailed unpaired *t* test with Pof8-myc. **D** Telomerase assay of myc-tagged Pof8 in a wild-type and *bmc1Δ* strain. A $^{32}$P-labeled 100-mer oligonucleotide was used as a loading control. Telomerase extension products were compared to a terminal transferase ladder, with +1 and +4 extension products indicated. Western blot probing for myc following myc immunoprecipitation is shown in the panel below. **E**, **F** Relative telomerase extension activity for myc-tagged Pof8 immunoprecipitates in a wild-type and *bmc1Δ* strain. The intensity of the +4 extension product was normalized to a precipitation loading control, then further normalized to TER1 expression (**E**) or the amount of TER1 immunoprecipitated with Pof8 (**F**). *P* values over each bar represent the results of a two-tailed unpaired *t* test with Pof8-myc ($n = 4$ biological replicates, mean ± standard error, *$P < 0.05$, **$P < 0.01$, ***$P < 0.001$, and ****$P < 0.0001$). Source data are provided as a Source Data file.

by Southern blot[50]. As expected, a strain lacking TER1 showed a complete loss of telomeric DNA, whereas deletion of *pof8* and *bmc1* only resulted in shorter telomeres, suggesting that while the two proteins are important for telomere maintenance, they are not strictly required like TER1 (Supplementary Fig. 6).

To better understand how Bmc1 affects telomere length and TER1 accumulation, we examined how the Pof8–TER1 interaction changes upon *bmc1* deletion. Somewhat surprisingly, considering that Bmc1 completely relies on Pof8 for recruitment to telomerase, we found that *bmc1* deletion also impaired Pof8 binding to TER1 (Fig. 5C), much like the cooperative binding of Pof8 and Lsm2-8 to TER1[16]. Similarly, we observed decreased telomerase activity immunoprecipitated by Pof8 in the *bmc1Δ* strain, with normalization of the intensity of the extension bands to TER1 levels revealing that the compromised activity cannot only be attributed to the loss of TER1 following *bmc1* deletion (Fig. 5D, E). Conversely, normalization of telomere extension activity to the amount of TER1 immunoprecipitated by Pof8 revealed that although less Pof8-containing telomerase complexes form in the absence of Bmc1, the ones that do form are not defective (Fig. 5D, F). Thus it appears that not only does Bmc1 affect TER1 accumulation but also contributes to Pof8 association with TER1, suggesting a further role for Bmc1 in ensuring holoenzyme functionality.

**Pof8-like proteins are associated with Bin3/Bmc1-like proteins in diverse fungal lineages.** The interaction between Bmc1 and Pof8, coupled with the conservation of such an interaction in other examined eukaryotes, led us to wonder at the extent of this interaction on a phylogenetic scale. While Bin3/Bmc1/MePCE (referred to in this section as Bin3 for ease of identifying fungal homologs) and LARP7/Pof8 family members are ubiquitous in higher eukaryotes, their absence or presence is more varied in fungal lineages. We queried hundreds of representative fungal species for the presence of Bin3- and Pof8-like proteins and identified many species with (a) both a Bin3 and Pof8 homolog, (b) only a Bin3 homolog, or (c) neither a Bin3 nor a Pof8 homolog (Fig. 6 and Supplementary Fig. 7). Conversely, only four out of 472 examined species contained a Pof8-like protein but no Bin3-like protein, and in these species the Pof8 homolog was noted to have a shortened N-terminal domain that would lack the La module observed in other LARP7-family members, raising the possibility that members of this rare cohort may not be *bona fide* LARP7-family members, at least as these are currently appreciated[20]. Based on the observed distribution, it seems likely that both genes were present in a fungi common ancestor, but Pof8 or both Bin3 and Pof8 were lost in certain lineages. Examination of the distribution of Bin3- and LARP7-family members in basal eukaryotic lineages reveals a similar pattern:

Bin3 can be present without LARP7, but there are no instances where a LARP7-family member exists in a lineage lacking a Bin3 homolog (Supplementary Table 2). Similarly, LARP7-family proteins are represented in Alveolates, Stramenopiles, Amoebozoa, Fungi, and Metazoans, while Bin3-family proteins are present in all eukaryotic lineages. Since the presence of Pof8/LARP7 is very highly linked to the presence of a Bin3/Bmc1 homolog, these data suggest that the function of LARP7-family members may be more intimately associated with the function of Bin3/Bmc1/MePCE than has been previously appreciated.

## Discussion

In this study, we have identified Bmc1 as a component of the *S. pombe* telomerase holoenzyme. In addition to showing an interaction between Bmc1 and U6 snRNA, we demonstrate that Bmc1 interacts with the telomerase RNA TER1 and components of the active telomerase holoenzyme in a manner that is dependent on the presence of the LARP7-family protein Pof8. Together, our results indicating that Bmc1 interacts with the mature, spliced form of TER1 and that Bmc1 immunoprecipitates possess telomerase activity in vitro strongly suggest that Bmc1 is a constitutive component of the telomerase holoenzyme containing Trt1, Pof8, Lsm2-8, and Est1 scaffolded on TER1 (Fig. 4E). Our results are in agreement with recent findings from the Baumann lab reporting the presence of Bmc1/Bin3 in the telomerase holoenzyme[41]. The idea of Pof8 and Bmc1 scaffolded on TER1 is reminiscent of LARP7 and MePCE in the 7SK snRNP, with Pof8/LARP7 recruiting Bmc1/MePCE to the RNA substrate. Similar to our data showing Pof8 recruiting Bmc1 to TER1 and the telomerase holoenzyme, LARP7 has also been demonstrated to recruit MePCE to the 7SK snRNP in human cells[24]. Thus, the protein–protein interaction and subsequent recruitment to RNA substrates by LARP7 appears evolutionarily conserved in fission yeast and higher eukaryotes. We also present data demonstrating that the presence of Pof8-like proteins is nearly universally correlated with the presence of Bin3-like proteins in fungi, suggesting that the Bmc1–Pof8 interaction is highly conserved and more prevalent than previously anticipated. Since Pof8 is required to load Bmc1 onto the telomerase RNA and U6 snRNA in fission yeast, this phylogenetic distribution raises the question of whether Bin3/Bmc1 homologs in fungi lacking a LARP7 homolog have evolved a mechanism to bind RNA targets in a Pof8/LARP7-independent fashion.

Since TER1 undergoes several processing steps before assembly into the telomerase holoenzyme, we were able to assess TER1 processing state to determine when Bmc1 interacts with TER1 with respect to the timing of TER1 maturation. We show that Bmc1-associated TER1 has a mature 3' end, indicative of spliceosome-catalyzed end processing, and a 5' TMG cap. This is

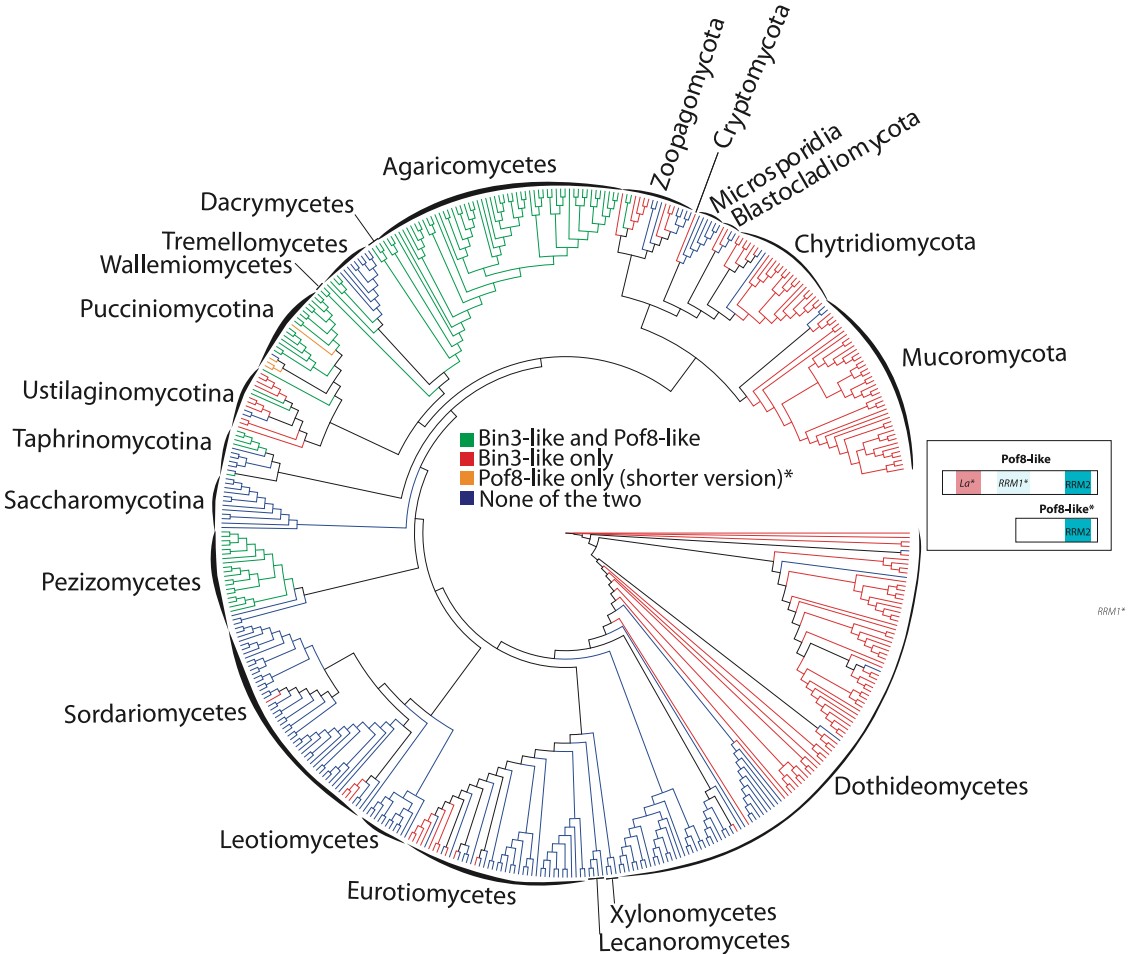

**Fig. 6 Phylogenic distribution of Bmc1/Bin3 and Pof8 in fungi.** Consensus cladogram describing the phylogenic relationships of 472 species representative of fungi phylum and classes and highlighting (using a color code) the distribution of Bin3 and Pof8 in these species. The cladogram is a consensus tree of 5328 distinct protein coding gene trees resulting from a genome-wide, against all, protein comparison (see "Methods"). Only posterior probabilities inferior to 1 are shown. The Bin3 and Pof8 distribution is recapitulated in Supplemental Data 3 with corresponding protein sequences. A cartoon presenting structural domains of Pof8 is presented. Pof8 La module and RRM1 are only inferred from secondary structure predictions and are represented by La* and RRM1* in the corresponding cartoon. In four species (colored in orange in the cladogram), only an N-terminal truncated version of Pof8 (Pof8*) that cannot accommodate the La module is present. Full cladogram with species names and statistical supports of the different nodes is presented in Supplementary Fig. 7.

consistent with previously reported data on the timing of Pof8 binding; since our data indicate that the interaction between Bmc1 and TER1 is dependent on Pof8, the most parsimonious explanation is that Pof8, and thus Bmc1, bind TER1 following 3' maturation and TMG capping, preceding the binding of the Lsm2-8 complex[13]. We show that the interaction between Bmc1 and TER1 is dependent on Pof8, leading to the conclusion that Bmc1 interacts with TER1 at the same stage as Pof8, resulting in the hierarchical assembly of the active telomerase holoenzyme.

The finding that Bmc1-associated TER1 is immunoprecipitated by a TMG antibody argues against a catalytic role for Bmc1 in TER1 processing. Our data, coupled with previous work characterizing the cap structure on the mature TER1 bound to Trt1[3,4], supports the idea that Bmc1 has a capping-independent function in the telomerase holoenzyme. LARP7 binding to MePCE in the 7SK snRNP inhibits its capping activity; MePCE, in turn, promotes the interaction between 7SK and LARP7, thereby stabilizing the complex[34]. In the *S. pombe* telomerase holoenzyme and U6 snRNP, we can hypothesize that Pof8 may also inhibit the methyltransferase activity of Bmc1 by binding to and occluding the active site. This is consistent with our results suggesting a transient interaction between Pof8 and U6 that may function to

load Bmc1 on U6. In such a model, an interaction between Bmc1 and Pof8 is necessary to recruit Bmc1 to U6 and following dissociation of Pof8, the catalytic site of Bmc1 would become available to cap U6. It is curious to note that *bmc1* deletion has no effect on U6 levels (Fig. 5A), although such a finding is consistent with what has been observed for human U6 upon MePCE depletion[26]. Since the lack of a MePCE-catalyzed cap on human 7SK is associated with an increase in exonucleolytic degradation[26], this therefore raises the question as to the function of the γ-monomethyl phosphate cap on U6, if not to protect the 5' end from exonucleolytic degradation.

In addition, our work identifying Bmc1 as a telomerase-associated protein brings to mind work identifying a box H/ACA motif at the 3' end of vertebrate telomerase RNA[51]. Box H/ACA RNAs form an RNP complex with Dyskerin/Cbf5, Nop10, Nhp2, and Gar1 to guide noncoding RNA pseudouridylation[52,53]. Although Dyskerin is a pseudouridine synthase, to date no evidence has pointed to a modification function for Dyskerin in the context of telomerase RNA. Rather, the Dyskerin-containing H/ACA complex has a structural role in telomerase biogenesis, with binding of the H/ACA RNP to telomerase RNA leading to a conformational change in the RNA that promotes 3' processing

and maturation over exosome-mediated decay[54]. This is reminiscent of the apparent catalytic independent role we observed for Bmc1 in fission yeast telomerase, although our data point toward Bmc1 having a role in promoting Pof8 binding to TER1, perhaps also through Bmc1-mediated conformational changes in TER1. The decreased TER1 levels observed upon *bmc1* deletion (Fig. 5A) can likely be attributed to the decreased interaction between Pof8 and TER1 in the absence of Bmc1. We therefore propose a model in which the direct interaction between Bmc1 and Pof8 both recruits Bmc1 to the telomerase holoenzyme, as well as promotes Pof8 binding to TER1. This in turn promotes Lsm2-8 binding to the exposed 3' end of TER1, which protects TER1 from exosome-mediated degradation[13,15,16]. The resulting TER1-containing RNP then recruits Trt1, forming a stable complex capable of extending telomeres.

Another facet of this work builds on the idea that components of the telomerase holoenzyme are more conserved than previously appreciated. In addition to the ubiquitous telomerase reverse transcriptase, previous work has identified LARP7-family proteins in both ciliates and yeast[11,13–15], arguing a level of evolutionary conservation. We now also provide evidence indicating the presence of RNase P and RNase MRP subunits in fission yeast, much like the RNase P and RNase MRP subunits found in the budding yeast telomerase complex[38–40]. The presence of RNase P and RNase MRP subunits in two divergent yeast species supports the importance of these factors in the function of a breadth of telomerase holoenzymes.

Our results indicating that Bmc1 is part of the telomerase holoenzyme were quite unexpected, as Bmc1 has been well-characterized in higher eukaryotes for its role in the processing, maturation, and protection of RNA polymerase III transcripts[26,34]. An intriguing idea involves polymerase switching through telomerase RNA evolution. Ciliate telomerase RNA is transcribed by RNA polymerase III and subsequently binds the LARP7 homolog p65 through the terminal polyuridylate stretch common to RNA polymerase III transcripts[11,17]. Since MePCE and LARP7 also associate with RNA polymerase III transcripts, it is possible that the Pof8-bound RNA polymerase II transcribed TER1, and the *S. pombe* telomerase RNP as a whole, represent an intermediate step in the evolution of telomerase RNA between ciliates and the RNA polymerase II transcribed telomerase RNA of higher eukaryotes. Future work should investigate the presence of Bmc1/MePCE homologs in the telomerase holoenzymes of other species, particularly in ciliates, which possess both an RNA polymerase III-transcribed telomerase RNA and a LARP7 homolog. Such findings will continue to develop emerging ideas regarding conservation in telomerase RNA processing and the composition of telomerase RNPs.

## Methods

**Strains, constructs, and growth media**. Strains were grown in yeast extract with supplements (YES) or Edinburgh Minimal Media (EMM), as indicated. Tag integration and knockouts were generated with a previously described PCR-based strategy and verified by PCR and western blotting[55] (primer sequences in Supplementary Table 3). Protein A-tagged strains were generated according to ref. [56]. The *bmc1Δ* strain was constructed by replacing the *bmc1* open reading frame with the phleomycin resistance cassette and flanking primers containing 750 nucleotides of homology to the *bmc1* genomic locus. Correct genotypes were selected on YES plates with the corresponding antibiotic (200 µg/mL G418, Sigma; 100 µg/mL Nourseothricin, GoldBio; 100 µg/mL phleomycin, Invivogen). Other strains were created by mating and antibiotic selection. A list of strains is provided in Supplementary Table 4.

**Native protein extract and immunoprecipitation**. *S. pombe* cells were grown in YES at 32 °C to mid-log phase, harvested, and subject to cryogenic disruption using a mortar and pestle. Cell powder was lysed in 50 mM NaCl, 20 mM Hepes pH 7.4, 55 mM KOAc, 0.5% Triton X, 0.1% Tween-20, 0.2 mM PMSF, 1:100 Protease Inhibitor Cocktail (Thermo, 78430), and 0.004 U/µL RNase inhibitor (Invitrogen, AM2694). For Protein A-tagged strains, immunoprecipitation was carried out with

Rabbit IgG-conjugated (MP-Biomedicals, SKU 085594) Dynabeads (Invitrogen, 14301) as described[57]. Myc-tagged proteins were immunoprecipitated with Protein G Dynabeads (Invitrogen, 10003D) coated with anti-myc antibody (Cell Signaling, 2276S). Beads were washed four times with 400 µL lysis buffer. For RNA immunoprecipitations, bound RNA was isolated by treatment of beads with 0.1% SDS and 0.2 mg/mL Proteinase K (Sigma, P2308) at 37 °C for 30 min, followed by extraction with phenol: chloroform:isoamyl alcohol (25:24:1) and ethanol precipitation.

**RNA preparation and northern blotting**. Total RNA was isolated from 1% of native protein extracts by incubation with 0.5% SDS, 0.2 mg/mL proteinase K (Sigma, P2308), 20 mM Tris HCl pH 7.4, and 10 mM EDTA pH 8.0 for 15 min at 50 °C, followed by phenol: chloroform extraction and ethanol precipitation. Northern blot analysis was performed as described using 8% TBE-urea polyacrylamide gels[58]. Briefly, electrophoresed RNA was transferred to positively charged nylon membranes (Perkin Elmer, NEF988001) with the iBlot 2 system (Thermo, IB21001) and probed with $^{32}$P γ-ATP-labeled DNA probes (probe sequences provided in Supplementary Table 5). For RNase H digestion, RNA was incubated with 25 pmol RNase H probe (provided in Supplementary Table 5) for 5 min at 65 °C and slowly cooled to 37 °C, followed by digestion with 5 U RNase H (NEB, M02975) for 30 min at 37 °C. The reaction was stopped with the addition of 25 µM EDTA pH 8.0 and phenol: chloroform extraction and ethanol precipitation. TMG-capped RNAs were isolated from RNase H-treated immunoprecipitates with Protein G Dynabeads coated with an α-TMG antibody (Sigma, MABE302), according to ref. [59].

**qRT-PCR and semi-quantitative RT-PCR**. 1 µg of Turbo DNase-treated RNA was reverse transcribed with the iScript cDNA synthesis kit (BioRad, 1708890), treated with 0.5 µL RNase cocktail (Invitrogen, AM2286), and diluted 1:10. cDNA was quantified using the SensiFAST SYBR No-Rox kit (Bioline, BIO-98005) and 1 µM of each primer (primer sequences in Supplementary Table 5). qPCR settings were as follows: 95 °C for 10 min and 40 cycles consisting of 10 s at 95 °C, 20 s at 60 °C, and 20 s at 72 °C, followed by melting curve analysis. TER1 and U6 levels were normalized to *act1* mRNA levels and the average wild-type Ct value, and subsequently subject to unpaired two-tailed Student's *t* tests and, where applicable, one-way ANOVA followed by a Tukey post hoc test with α set to 0.05 (Supplementary Data 4).

For semi-quantitative RT-PCR, DNase-treated RNA, 10 nmol dNTP mix, and 10 pmol gene-specific reverse primers (Supplementary Table 5) were heated to 65 °C and slow-cooled to 37 °C before reverse transcription with 5 U AMV-RT (NEB, M0277L) at 42 °C for 1 h. cDNA was amplified with Taq polymerase (NEB, MO273L) using standard protocols and the following cycling conditions: 5 min initial denaturation at 94 °C, 22 (TER1) or 17 (U6) cycles of 30 s at 94 °C, 30 s at 57 °C, and 1 min at 72 °C, and a final 10 min extension at 72 °C.

**RIP-Seq**. Libraries were constructed from immunoprecipitated RNA samples by the RNomics Platform at the Université de Sherbrooke in Sherbrooke, Quebec. RNA quality was assessed with a Bioanalyzer small RNA chip (Agilent, 5067-1548). Libraries were constructed with the NEBNext Ultra II Directional Kit (NEB, E7760S) and amplified with ten PCR cycles. cDNA libraries were sequenced on an Illumina NextSeq 500 with two runs per sample, each for 50-bp single-end reads. Following fastp processing (Version 0.20.1), reads from the first sequencing run were aligned to the fission yeast genome (ASM294v2) with Bowtie 2[60,61] and counted with featurecounts[62] using the EF2 build of the ENSEMBL fission yeast genome. Differential expression analysis was performed using edgeR, with reads filtered to include transcripts with at least 1 count per million (CPM) in each sample, and libraries were normalized by Trimmed Mean of M-values (TMM)[63,64].

**Circularized rapid amplification of cDNA Ends (cRACE)**. Immunoprecipitated RNA was treated with 2 U TURBO DNase (Invitrogen, AM2239) and dephosphorylated with 1 U calf intestinal alkaline phosphatase (NEB, M0290), followed by decapping and circularization with RNA 5' Pyrophosphohydrolase (NEB, M0356S) and T4 RNA Ligase 1 (NEB, M0204S). TER1 was amplified from circularized RNA with the OneStep RT-PCR Kit (Qiagen, 210210) (primer sequences in Supplementary Table 5). cRACE products were cloned into the pGEM-T vector (Promega, A1360) and sequenced by the TCAG DNA Sequencing Facility at the Hospital for Sick Children in Toronto.

**Mass spectrometry and GO analysis**. Half of the native protein extract was pre-treated with 0.625 U/µL Benzonase (Millipore, E1014) at 37 °C for 30 min. The reaction was stopped with 5 mM EDTA pH 8.0 and Protein A immunoprecipitation was performed in the same manner as for RNA immunoprecipitations. After washes with lysis buffer, beads were washed with 0.1 M NH$_4$OAc and 0.1 mM MgCl$_2$ and eluted with 0.5 M NH$_4$OH for 20 min at room temperature. Eluates were lyophilized and subject to in-solution tryptic digestion. LC-MS/MS analysis was performed by the IRCM Proteomics Discovery Platform on a Q Exactive HF.

Raw data were processed and analyzed using the MaxQuant software package 1.5.1.2[65] and the fission yeast UP000002485 reference proteome (24/11/2019). Settings used for MaxQuant analysis were as reported[66]. Proteins present in

immunoprecipitations from both biological replicates and absent from control immunoprecipitations were considered genuine Bmc1-interacting partners and ranked by peak intensity. Proteins that showed a complete loss of spectral counts upon benzonase addition were considered nucleic acid-dependent interactions. PANTHER overrepresentation tests (GO Ontology database released 2019-12-09) were conducted on the top 50 Bmc1-interacting partners. Fisher's exact tests with false discovery rate (FDR) correction were performed using the GO biological process complete and GO cellular component complete datasets.

**Co-immunoprecipitation and western blotting**. S. pombe cells were grown in YES at 32 °C to mid-log phase, harvested, and subject to cryogenic disruption using a mortar and pestle. Cell power was lysed in Co-IP lysis buffer (10 mM Tris HCl pH 7.5, 150 mM NaCl, 0.5% NP40, 1:100 Protease Inhibitor Cocktail). Immunoprecipitations were carried out as described and proteins were eluted by resuspending beads in 2× SDS loading buffer (100 mM Tris HCl pH 6.8, 4% SDS, 20% glycerol, 0.2% bromophenol blue) and boiling at 95 °C for 5 min. Western blot analyses were performed using monoclonal anti-myc (Cell Signaling, 2276 S) at 1:5000 and monoclonal anti-beta-actin (Abcam, ab8226) at 1:2500 for primary antibodies and HRP-conjugated anti-mouse (Cell Signaling, 7076) at 1:5000 for secondary antibodies. Protein A-tagged proteins were detected with HRP-conjugated polyclonal anti-Protein A (Invitrogen, PA1-26853) at 1:5000.

**Glycerol-gradient sedimentation**. Native protein extracts were separated on a 10 mL 20–50% glycerol gradient (20 mM HEPES pH 7.6, 1 mM MgOAc, 1 mM DTT, 300 mM KOAc) and spun in an SW41 rotor for 20 h at 30,000 rpm according to ref. [67]. Individual fractions were divided into two for protein extraction by TCA precipitation and RNA extraction with phenol:chloroform.

**Telomerase activity assay**. The telomerase activity assay was performed as described[13]. Briefly, bead slurries (see immunoprecipitation) were incubated at 30 °C for 90 min in a 10 µL reaction containing 50 mM Tris HCl pH 8.0, 1 mM MgOAc, 5% glycerol, 1 mM spermidine, 1 mM DTT, 100 mM KOAc, 0.2 mM dATP, dCTP, and dTTP, 5 µM telomerase assay primer (Supplementary Table 5), and 0.3 µM 3000 Ci/mmol [α-$^{32}$P] dGTP (Perkin Elmer, BLU512H250UC). The reaction was stopped with the addition of 0.5% SDS, 10 mM EDTA pH 8.0, 20 mM Tris HCl pH 7.5, 2 µg/µL Proteinase K (Sigma), and 1000 cpm [γ-$^{32}$P] ATP-labeled 100-mer oligonucleotide and incubation at 42 °C for 15 min, followed by phenol:chloroform extraction and ethanol precipitation. RNase A-treated samples were pre-incubated with 20 ng RNase A (Invitrogen, AM2271) at 30 °C for 10 min. Extension products were separated on a 10% urea-TBE polyacrylamide sequencing gel at 60 W for 90 min, dried and exposed to a PhosphorScreen, and imaged with a Typhoon imager.

**Genomic DNA extraction, southern blotting, and chromosome fusion PCR**. Genomic DNA was extracted from logarithmically growing cells according to ref. [14]. For southern blotting, 15 µg genomic DNA was digested overnight with EcoRI or ApaI and separated on a 1% agarose gel, transferred to positively charged nylon membranes (Perkin Elmer, NEF988001) by capillary transfer in 10× SSC, and probed with a $^{32}$P γ-ATP-labeled telomere probe (probe sequence provided in Supplementary Table 5). The chromosome fusion PCR was adapted from[13]. Briefly, 50 µL PCR reactions contained 1 µL genomic DNA, 0.4 µM forward and reverse primers (sequences provided in Supplementary Table 5), 200 µM dNTPs, 1× Taq ThermoPol Buffer (NEB), and 1.25 U Taq polymerase (NEB, M0273). The Trt1 gene was amplified as a loading control in an identical PCR reaction containing the corresponding primers. The PCR reaction consisted of an initial denaturation at 95 °C for 5 min, followed by 32 cycles of 95 °C for 15 s, 55 °C for 30 s, and 68 °C for 3 min, with a final extension at 68 °C for 10 min.

**Structure prediction**. The structure of S. pombe Bmc1 was predicted using Phyre2[68] and aligned with the co-crystal structure of the H. sapiens MePCE methyltransferase domain bound to 7SK snRNA and SAH (6DCB[31]) in PyMol[69]. Primary sequences were aligned with the Clustal Omega Multiple Sequence Alignment tool[70].

**Phylogenetic tree construction and search for Bin3-like and Pof8-like sequences**. Protein sequences (GeneCatalogs) of 472 fungal species were downloaded from the JGI website (https://mycocosm.jgi.doe.gov/) in fasta format. In total, 5,614,520 proteins were downloaded and merged into a single multifasta file. A blast (Blastp) against all proteins was performed using diamond blastp[71] with a minimum e-value of 1e-5 and outfmt 6 (for tabular output). In order to identify homologous proteins, proteins were clustered together if they shared at least 50% similarity over 70% of coverage using SiLiX[72]. When a cluster contained two or more paralogous genes derived from duplications, the species were removed from the cluster. Finally, only clusters with a minimum of 100 species were retained for further analysis. Following these cleaning steps, we obtained 5328 clusters containing one gene per species. Proteins sequences of each cluster were aligned using MAFFT (v7.271) with default parameters (https://mafft.cbrc.jp/alignment/server/). Multiple alignments were cleaned using trimA1 (v1.4.rev22 build [2015-05-21]) with -st 0.01 option. Fastree (version 2.1.10 SSE3) with default parameters were used to build phylogenetic trees[73]. The consensus fungi tree was obtained using

ASTRAL (astral.5.6.3) software with default parameters[74]. To detect the presence of genes coding for Bin3-like and Pof8-like proteins, blast searches (blastp and tbalstn) were performed on each studied fungus genome, using initially either the full-length S. pombe Bin3/Bmc1 protein sequence or the S. pombe Pof8 xRRM conserved motif. This procedure was repeated in each fungal clade using clade-specific Bin3-like and Pof8-like sequences detected by the initial search using S. pombe sequences. Finally, new blast searches were performed all over again, this time using Bin3-like or Pof8-like "probes" made of a collection of sequences representing the diversity of these two proteins in fungi.

**Statistics and reproducibility**. For qPCR analyses, two-tailed unpaired Student's t tests were performed and, where applicable, one-way ANOVA followed by a Tukey post hoc test with a set to 0.05. RIP-Seq, RNA IP coupled to qPCR, and total RNA qPCR experiments were performed in biological triplicate, IP mass spectrometry, IP western blots, RNA IP coupled to northern blots, glycerol-gradient analysis, and telomere southern blots were repeated in biological duplicate, and telomerase assays were repeated in biological quadruplicate.

**Reporting summary**. Further information on research design is available in the Nature Research Reporting Summary linked to this article.

## Data availability

The data supporting the findings of this study are available from the corresponding authors upon reasonable request. The mass spectrometry proteomics data have been deposited to the ProteomeXchange Consortium via the PRIDE[75] partner repository with the dataset identifier PDX023356 and 10.6019/PDX023356. RNA-Seq data have been deposited in NCBI's Sequence Read Archive (SRA) database under BioProject number PRJNA776661. Source data are provided with this paper.

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

## Acknowledgements

We thank Toru Nakamura and Virginia Zakian for sharing published yeast strains and Marc Fabian and Raymund Wellinger for helpful comments on the manuscript. J.P. is supported by a Canada Graduate Scholarship (Doctoral) from the National Sciences and Engineering Research Council of Canada. M.A.B. is supported by a Discovery Grant from NSERC ("Impact of chemical modification of noncoding RNAs on gene expression in *S. pombe*"). J.M.D. is supported by the Institut Universitaire de France (IUF), J.-M.D. and M.E.B. by the CNRS and the University of Perpignan (UPVD) and their study is set within the framework of the "Laboratoires d'Excellences (LABEX)" TULIP (ANR-10-LABX-41) and of the "École Universitaire de Recherche (EUR)" TULIP-GS (ANR-18-EURE-0019).

## Author contributions

J.P. conceived of the study, designed, performed, and analyzed all experiments, and wrote the first draft of the manuscript with input from all authors. M.E.B. and J.-M.D. performed phylogenetic analyses. J.P. analyzed RNA-Seq data with help from J.G. M.A.B. conceived of the study, designed experiments, supervised the project, and edited the manuscript.

## Competing interests

The authors declare no competing interests.
