## [Peer Review File · Nature Communications]

Title: The methyl phosphate capping enzyme Bmc1/Bin3 is a stable component of the fission yeast telomerase holoenzymeREVIEWER COMMENTS

Reviewer #1 (Remarks to the Author):

In their study Porat et al. identify Bin3, the *S. pombe* homolog of MePCE, as novel component of the telomerase holoenzyme. They found that Bin3 is recruited to telomerase through protein contacts with Pof8, a homolog of LARP7, which has recently been identified as telomerase component. The interaction between Bin3/MePCE and Pof8/LARP7 thus appears to be evolutionarily conserved. Through a series of experiments the authors show that Bin3 associates with telomerase containing mature TER1 and that Bin3 is a negative regulator of telomerase levels and activity. The research is novel and extends the current understanding of *S. pombe* telomerase. The manuscript is well-written and logically structured.

Major comments:

The major limitation of the study by Porat et al. is the lack of mechanistic insights into the regulatory role of Bin3 for telomerase assembly and function. The authors found that depletion of Bin3 increases TER1 levels resulting in increased telomerase activity. But how Bin3 exerts this inhibitory function remains unclear. For example, does Bin3 affect the function of Pof8 in Lsm2-8 deposition? Does it affect the structural stability of the telomerase complex or activate degradation pathways? Experimental evidence investigating these (or other) possibilities would greatly strengthen the manuscript.

Minor comments:

Fig. 1C: there is no clear Northern blot band for TER1 in the input and the molecular weight marker is not included. The authors might also consider using qPCR for quantitative detection of TER1 in the Bin3 PrA immunoprecipitate.

Fig. 2C: there is a U6 band present for the TMG IP. Is it possible that the IP conditions were not stringent enough?

Fig. 3B and C: some of the Western blots appear overly processed in terms of brightness/contrast.

Fig. 3: have the authors considered additional means to validate the interaction of Bin3 with the telomerase complex? For example, size-separation of *S. pombe* lysate by sucrose gradient ultracentrifugation followed by Western blot detection of telomerase components and Bin3 would give additional evidence that Bin3 associates with the telomerase complex.

Fig. 4A: there is no TER1 Northern blot signal in the inputs.

Fig. 5A and B: statistical analyses should be performed with one-way ANOVA and post hoc test.

Reviewer #2 (Remarks to the Author):

In the current manuscript, Porat et al. provide evidence that fission yeast Bin3 interacts with fission yeast telomerase via LARP7 protein Pof8. While I do find their overall findings to be intriguing and exciting, I found multiple issues with the current study that I would like them to address as listed below. Overall, I find evidence that Bin3 interacts with an active telomerase complex via Pof8 to be convincing. However, I do not believe that authors convincingly established functional significance of Bin3-telomerase interaction, as I found evidence that Bin3 negatively regulate telomerase-dependent telomere elongation from Southern blot analysis (Fig.5) to be preliminary and lack critical controls.

Experimental and data interpretation issues:

1. Supplemental Figure 1C and 1D are lacking control cells to fully evaluate how wt or bin3Δ cells carrying GST-Bin3 plasmid grow in the presence or absence 5FOA or thiamine (B1). It appears that wt cells carrying GST-Bin3 plasmid grow much worse than bin3Δ cells with the plasmid? It would be more useful and easier to compare growth behavior if serial dilution spots for (1) wt+empty plasmid, (2) wt+GST-Bin3 plasmid, and (3) bin3Δ+GST-Bin3 plasmid are plated on the same plate side-by-side on (1) EMM-ura, (2) EMM-ura+B1, or (3) EMM+5-FOA. As it is shown, it is not even clear if panels shown on supplemental figure 1D come from the same plate or pictures taken on the same day for different strains. I am also not certain the rationale of expressing GST-tagged Bin3, rather than non-tagged version of Bin3, as any phenotype associated with overexpression could be attributable to expression of mutant (i.e. GST-tagged) Bin3, rather than simply increasing expression of wt Bin3 protein.

2. RIP-Seq data presented in supplemental Table for Bin3 seems to be wildly different for 2 replicates for Bin3-PrA IP. In one experiment, Ter1 RNA (SPNCRNA.214.1) and U6 RNA are indeed highly represented, while 2nd experiment found Ter1 or U6 to be barely above background non-tag control. It seems to me that enrichment of Ter1 or U6 would not be counted as significant in replicate 2 if this result is independently analyzed. However, by taking average values of 2 experiments, Figure 1A would give false impression that U6 and Ter1 are reproducibly identified in both replicates of RIP-Seq experiments. Thus, while authors provide convincing follow-up data to substantiate the notion that Bin3 do indeed interact with U6 and Ter1 RNAs, I would encourage authors to plot data from replicate 1 and replicate 2 of Bin3 RIP separately (at least in supplemental data). It is more honest reflection of actual data, and perhaps authors can speculate why 2nd replicate did not work well, but results from replicate 1 where very high enrichment of U6 and Ter1 were observed in RIP analysis could still be used to rationalize more detailed follow-up experiments described in the manuscript.

I am also not entirely sure what exactly is plotted in Figure 1A as I don't quite understand labels for X- and Y-axis. What exactly are $\text{Log}_2(\text{Bin3 IP} + 2)$ and $\text{Log}_2(\text{Control IP} + 2)$ represent? Is there a significance to "+2"? Perhaps authors can include actual Log_2 values used in making Figure 1A plot in supplemental data table (and leave in formula for generating such values in Excel sheet) so that it is more obvious how data is analyzed.

Supplemental Table for RIP-Seq is also difficult to understand and not very accessible. Rather than only listing systematic name, common/functional names of RNA should be given where such names are available. For example, SPNCRNA.214.1 should be made clear that it is Ter1, while SPSNRNA.06.01 should also be listed as “snu6 / small nuclear RNA U6”, according to the description given in Pombase. Likewise, snoRNA, tRNA, rRNA, etc. should be clearly identified. There are also large number of “ENSRNA” listed in this table. I am not familiar with such RNA, and I would appreciate some explanation. Are ENSRNAs also included in Figure 1A as well, or only well-defined RNA? The supplemental table also includes some “dubious” RNA (for example SPCC417.15.1) according to Pombase. Are such RNAs also included (or excluded) in Figure 1A? I would also suggest replacing 25_yAS99, 26_yAS99, 31_Bin3, 32_Bin3, etc. with something more descriptive such as no tag control IP, Bin3 IP, etc. in supplemental Table 1.

3. Lines 172-3: While Bin3-associated U6 snRNA was not immunoprecipitated by the anti-TMG antibody... / I find this statement misleading as U6 signal is clearly visible in TMG IP sample. While it is true that anti-TMG seems to be less efficient in depleting U6 RNA compared to U5 RNA in flowthrough fraction, signal for U6 is only marginally weaker than U5 in TMG IP lane. Thus, authors cannot claim that anti-TMG antibody does not IP U6 RNA. It is either that anti-TMG antibody can cross-react to gamma-monomethyl phosphate cap OR there is a subfraction of U6 carrying in TMG cap, based only on data presented in Fig. 1C-D. If U6 RNA was previously established to be exclusively capped by monomethyl-cap, then it would suggest that anti-body is not as specific to TMG as suggested by authors? Furthermore, I think it would be too strong of a statement to suggest that “TER1 is not a substrate for Bin3-catalyzed modification” (Line 179) unless author can more robustly demonstrate complete absence of monomethyl-cap by utilizing antibody specific to monomethyl-cap or by some other sensitive method to detect monomethyl cap on RNA.

4. Supplementary Tables 2 & 3 are incomplete for systematic gene names for fission yeast as they only list cosmid names (which includes multiple genes) and do not indicate individual ORF. (For example, Bin3 should have been indicated as SPBC2A9.10, rather than SPBC2A9.) This omission significantly reduces usefulness of these tables. To make sure readers can clearly identify proteins that are found in mass spec. analysis, unique systematic names should be included for all genes, even for those that have been named. Furthermore, authors should update protein description for listed genes as some of them seems to be outdated. For example, "uncharacterized RNA binding protein SPBC1861" should have been listed as prp24 and SPBC1861.04c, according to the latest description given in Pombase. (I suppose the version of fission yeast protein database used in mass spec. analysis was somewhat outdated, and thus used old name/descriptions?)

5. Line 208: Authors mention that “the Bin3-Trt1 interaction is lost” upon treatment with benzonase. While I find that Bin3-Trt1 interaction is reduced in benzonase treated sample, residual interaction is

clearly visible. Thus, I would suggest that authors modify their statement to reflect only partial disruption of Bin3-Trt1 interaction is found in benzonase treated sample, as “lost” to me imply that the interaction is completely disrupted. Rather than treatment with benzonase, authors might also consider monitoring Bin3-Trt1 interaction in *ter1Δ* cell background, which might more completely disrupt Bin3-Trt1 interaction if Ter1 RNA is essential for mediating Bin3-Trt1 interaction.

6. Lines 229-231: Authors concluded that Bin3-Ter1 interaction is dependent on Pof8 based on loss of Ter1 signal for Bin3 IP in *pof8Δ* background. However, since previous studies have determined that *pof8Δ* cells show greatly reduced Ter1 RNA in cells, it is not entirely clear if northern blots shown in Fig. 4A is sensitive enough to detect Bin3-Ter1 interaction. It would also be good to determine what fraction of input RNA is pulled down in Bin3 IP in wt vs *pof8Δ* cells to control for greatly reduced Ter1 level in *pof8Δ* cells, but since input Ter1 show no signal, it might not be possible to determine such values by northern blots. Thus, it would be worth monitoring Bin3-Ter1 interaction by utilizing RT-PCR to determine efficiency of Ter1 IP for wt vs. *pof8Δ* cells.

7. Lines 255-256: Authors suggest that loss of telomerase activity in Bin3 pull down in *pof8Δ* cells support the notion that Pof8 is required for Bin3-telomerase interaction. I don't think this is a valid argument since Peter Baumann's group has shown that in vitro telomerase activity is barely detectable in *pof8Δ* cells even when Trt1 IP samples were used in assay. Since *pof8Δ* cells show greatly reduced active telomerase, it would have been nearly impossible to detect telomerase activity in *pof8Δ* cells even if Bin3 is still robustly interacting with Ter1.

8. Lines 265-268 & supplementary Fig. 4: Authors state that overexpression of Bin3 does not rescue telomere shortening and telomere fusions observed in *pof8Δ* cells, but only show PCR assays for detecting telomere fusion and no data on how Bin3 overexpression affects telomere length by Southern blot. I think authors should monitor telomere length before claiming that Bin3 over-expression does not rescue telomere shortening in *pof8Δ* cells.

On a minor note, plasmid name indicated in this figure does not match other figures. Does this figure utilize non-GST-tagged plasmid, or same pREP4x-GST-Bin3 plasmid as other figures? Strain list (supplementary Table 5) for this figure does not actually include plasmid, so I am not sure exactly what is shown in supplementary Fig. 4

9. Based on data shown in Fig. 5, authors argue that Bin3-telomerase interaction negatively regulates telomerase-dependent telomere elongation by negatively regulating TER1 accumulation in cells. However, I am not entirely convinced if claimed slight elongation of telomeres in cells expressing reduced level of GST-Bin3 is significant based on Southern blot presented in Fig. 5C. There are several issues/suggestions to improve robustness of Southern blots as listed below.

(a) Since average telomere length maintained by cells are very sensitive to growth condition and even wt cells show changes in average telomere length depending on media used to grow cells, temperature, etc. It is critical that authors include additional controls to ensure that growing cells in the presence (+B1) or absence (-B1) of thiamine did not cause telomere length change in control cells. Authors should minimally monitor telomere length for cells grown in same number of generations in same media with or without added thiamine (B1) for (1) wt cells carrying empty pREP41-GST plasmid, (2) wt cells carrying pREP41x-GST-Bin3, (3) bin3Δ cells carrying pREP41x-GST-Bin3. For a given genotype and growth condition, multiple independently derived clones should be utilized as well since plasmid copy number can vary greatly among independent transformants in fission yeast.

(b) Quality of gDNA used in Fig. 5C seems questionable as wt cell lane contain unexplained 2nd peak of telomere signal much smaller than 1 kb, which is not usually seen in Southern blot for EcoRI digested gDNA (Wt cells should show peak of telomere signal ~1 kb for telomeres.) I suspect this might be caused by degradation of gDNA. For unknown reason, when fission yeast cells are grown in minimum media, we have also seen appearance of these smaller band if culture is over-saturated. Thus, it would be worth repeating gDNA prep by ensuring that cells are still in log-phase. Another issue with gDNA prepared from cells grown in minimum media is that EcoRI digestion would frequently fail if starting concentration of gDNA is too low. Such incomplete digestion would often produce appearance of much larger product on Southern blot. For bin3Δ+4x GST-Bin3+B1 sample, there appears to be 2 distinct peaks with much bigger product is observed, and I suspect this is due to partial digestion of gDNA. To avoid partial digestion, authors should prepare gDNA samples from larger culture (and ensuring exponential cell growth when samples are collected).

(c) As I mentioned earlier when discussing results shown in supplementary Fig. 1, I am not entirely sure why authors are using GST-tagged Bin3 expression plasmid if they want to examine effect of reduction in Bin3 expression. I think that use of pREP41 plasmid without tag would be preferred to make sure that GST-Bin3 is not causing unexpected dominant phenotype. I also wonder if authors may be able to obtain more clear change in telomere length if they were to utilize pREP81 series plasmid, which should express even less Bin3. Did authors found that pREP81-Bin3 would not be allow bin3Δ cells to survive?

(d) I do see that perhaps very slight shift in peak of telomere signal that are found between 1 kb and 1.5 kb ladder. However, resolution of EcoRI digested Southern blot is not sufficient to see such small change in telomere length. To see slight changes in telomere length better, Apal digest should be used in Southern blot since average telomere length in Apal-digested Southern blot is much smaller (~300bp) and thus able to better resolve small changes in telomere length. Quantification of telomere peak would also be helpful.

(e) Labels for Fig.5 is confusing. For 5A and 5B, bin3Δ cells are presumably carrying pREP4x GST-Bin3 plasmid, but this is not indicated. For Fig.5C, I assume that wt and pof8Δ cells are carrying empty pREP4x-GST plasmid as both strains are indicated with "4x", but strain table (supplementary Table 5) does not indicate any plasmid for wt and pof8Δ cells. Do wt and pof8Δ cells carry plasmid or not?

10. Lines 339-340: As I discussed on point 6 above, it is not entirely clear if “interaction between Bin3 and TER1 is dependent on Pof8” since *pof8Δ* cells carry greatly reduced TER1 RNA and it is not clear Northern blot can detect reduced TER1 RNA in Bin3 pull down.

11. Lines 370-373: Authors seem to suggest that inability to limit telomerase activity in the absence of Bin3 might be the cause of unviability for *bin3Δ* cells since deletion of *tpz1* or *stn1* are also “essential” proteins. I don’t think this is a valid argument and it should be removed. First of all, “essential” function for *TPZ1* and *STN1* lies with telomere capping function, and second, viable *tpz1Δ* and *stn1Δ* cells can be obtained since survivor cells that carry circular chromosomes can be readily obtained in fission yeast. It seems to be that the actual cause of *bin3Δ* inviability probably lies with its function in regulation of U6 RNA as many of splicing factors are essential in fission yeast.

12. Lines 384-385: Authors mention the study that found reduced telomere length for LARP7 mutant patient to suggest that the role of LARP7 in telomerase regulation might be conserved in humans as well. However, the cited study found that LARP7 patient shows alteration in TERT splicing, which is likely the cause of the telomere phenotype. To date, human LARP7 has not been identified as an integral part of the telomerase holoenzyme complex, and recent studies have found functional roles for LARP7 in U6 RNA regulation, it seems rather unlikely that human LARP7 would play a conserved role in the telomerase holoenzyme complex.

Minor editorial issues:

1. For the introduction section, authors could improve the citation of previous research. Here are some examples. For line 30: 1st sentence could use some references. For line 35: also cite Nakamura et al. Science 19997 for *S. pombe* *Trt1*. For line 39: authors should cite primary papers (or reviews) which established that yeast and metazoan telomerase RNA is transcribed by RNA Pol II. For line 68: authors should include Juli Feigon papers where her group identified and described the xRRM motif.

2. Figure legend for Fig 1D should also mention U6 and U5.

3. Figure legend for Fig. 2B should also mention U6 data.

Reviewer #3 (Remarks to the Author):

In this manuscript, the Bayfield group investigates the biochemical composition of the *S. pombe* telomerase holoenzyme. They report on the unexpected observation that the capping enzyme Bin3 is a stable component of this Sm/Lsm type RNP. The authors further provide evidence that Bin3 associates apart from the U6snRNA also with the telomerase holoenzyme via an interaction with the protein Pof8. Importantly, this factor belongs to the Larp7 family, which has been linked to 7SK RNP and more recently also to U6 modification. The data clearly show that Bin3 does not generate the cap of TER1 but rather affects the composition, assembly, and regulation of the telomerase holoenzyme in yeast.

This is a potentially interesting study providing valuable new insight into telomerase biology and its evolution. The following aspects should be addressed to further support the conclusions drawn by the authors:

1. It is not clear to me whether the particle schematically shown in Fig. 4D exists with the indicated set of proteins or whether this is actually a mixture of particles with different protein compositions. If technically possible, the authors should add (biochemical) experiments that further illustrates the composition of the newly defined particle.
2. In general, I found the quality of several Northern and Western blots presented in the manuscript of low quality, which makes the interpretation of the data difficult. Sometimes, the bands are not even visible (see for example 1 B, 2D). The respective blots should be replaced. Input lanes often have a weaker or no signals than the IP lanes (see 1C, 4A. I assume that only a small fraction of the inputs is loaded, which should be mentioned). Fig. 2D is not mentioned in the text.
3. The enzymatic activity assayed in Fig. 4C is not properly controlled as it is not clear whether comparable amounts of material (Trt1, Pof8, Bin3) has been included in this assay. The quality of this experiment can also be improved as this is a well-established assay. The lower panel of this figure is not labelled.

Please find below our responses to reviewer comments for our manuscript “The methyl phosphate capping enzyme Bin3 is a stable component of the fission yeast telomerase holoenzyme” by Porat and coworkers. We have addressed all of their comments and are certain the manuscript has been substantially improved as a result, for which we thank them.

We address the concerns of the reviewers below:

Reviewer #1:

In their study Porat et al. identify Bin3, the S. pombe homolog of MePCE, as novel component of the telomerase holoenzyme. They found that Bin3 is recruited to telomerase through protein contacts with Pof8, a homolog of LARP7, which has recently been identified as telomerase component. The interaction between Bin3/MePCE and Pof8/LARP7 thus appears to be evolutionarily conserved. Through a series of experiments the authors show that Bin3 associates with telomerase containing mature TER1 and that Bin3 is a negative regulator of telomerase levels and activity. The research is novel and extends the current understanding of S. pombe telomerase. The manuscript is well-written and logically structured.

We were pleased that this reviewer had a positive impression of our work and thank them.

Major comments:

The major limitation of the study by Porat et al. is the lack of mechanistic insights into the regulatory role of Bin3 for telomerase assembly and function. The authors found that depletion of Bin3 increases TER1 levels resulting in increased telomerase activity. But how Bin3 exerts this inhibitory function remains unclear. For example, does Bin3 affect the function of Pof8 in Lsm2-8 deposition? Does it affect the structural stability of the telomerase complex or activate degradation pathways? Experimental evidence investigating these (or other) possibilities would greatly strengthen the manuscript.

Our revised manuscript contains substantial changes that provide greater mechanistic insight into the role of Bin3 in telomerase function. Much of this work relies on our newly successful construction of a Bin3 knockout strain. Existing literature is conflicting with respect to the essentiality of Bin3: the fission yeast genomic database Pombase reports that Bin3 is essential, although a genome-wide deletion library (Kim, D. U. *et al.* Analysis of a genome-wide set of gene deletions in the fission yeast *Schizosaccharomyces pombe*. *Nat. Biotechnol.* (2010) doi:10.1038/nbt.1628.) reported low confidence for their Bin3 knockout strain. Since our original submission, we obtained the diploid Bin3^{+/-} strain from the *S. pombe* genome deletion library (Bioneer), which also annotates Bin3 as essential, and discovered that the strain was not constructed correctly. We note that, according to the PomBase database, there are a number of other essential genes closely linked to the Bin3 locus, and we therefore hypothesize that various attempts to knock out this allele (including ours in our previous submission) may have resulted in unanticipated changes we did not previously detect to essential genes at other closely linked loci.

Due to this uncertainty, we reattempted this and have successfully generated a viable Bin3 knockout strain, using increased homology to the Bin3 locus to increase the efficiency and specificity of recombination. We have verified this new strain with PCR and sequencing (supplementary figure 5), as well as back-crossing it to a wild-type strain and monitoring lethality. Using this knockout strain, we performed additional experiments to add mechanistic insight into the role of Bin3 in telomerase function and now demonstrate in the new manuscript (Figure 5) that Bin3 deletion decreases steady-state TER1 levels, but not U6 snRNA levels. Bin3 deletion further impairs the recruitment of Pof8 to the remaining TER1, suggesting that Bin3 promotes hierarchical complex assembly and subsequent telomerase activity, which we confirm results in shorter telomeres. As Pof8 has been previously linked to protection of TER1 from nuclear exosome mediated (Rrp6 dependent) degradation, we also show that decreased TER1 levels in the dBin3 strain can be rescued by deletion of Rrp6. Thus, data in our revised manuscript are consistent with Bin3 cooperating with Pof8 to promote telomerase assembly and to protect TER1 from exosomal degradation.

Minor comments:

Fig. 1C: there is no clear Northern blot band for TER1 in the input and the molecular weight marker is not included. The authors might also consider using qPCR for quantitative detection of TER1 in the Bin3 PrA immunoprecipitate.

We have included a molecular weight marker and added an additional panel to figure 1C containing a semi-quantitative RT-PCR, which allowed for the detection of lowly expressed TER1 in inputs.

Fig. 2C: there is a U6 band present for the TMG IP. Is it possible that the IP conditions were not stringent enough?

We thank the reviewer for pointing this out. We switched from antibody-conjugated agarose beads to an antibody and magnetic beads, which allowed a pre-clearing step to reduce non-specific binding. We also considered that U6 may be indirectly pulled down with the TMG antibody due to an RNA-RNA interaction between U6 and another TMG-capped RNA (i.e. U4, see U4 enrichment highlighted in our RIP-Seq below). To address this, we also added an additional step to heat denature RNA prior to bead and antibody addition. As a result the amount of U6 enriched in the TMP IP is reduced to nearly undetectable levels (Figures 2C-D).

Fig. 3B and C: some of the Western blots appear overly processed in terms of brightness/contrast.

We have repeated these western blots and the new images are present in figures 3B and C.

*Fig. 3: have the authors considered additional means to validate the interaction of Bin3 with the telomerase complex? For example, size-separation of *S. pombe* lysate by sucrose gradient ultracentrifugation followed by Western blot detection of telomerase components and Bin3 would give additional evidence that Bin3 associates with the telomerase complex.*

We thank the reviewer for this insightful suggestion and have performed this experiment, now seen in Figure S3, where we see co-sedimentation of Bin3, Pof8, and TER1 in a glycerol gradient. Notably, we see a shift of Bin3 to lighter molecular weight fractions in the absence of Pof8, further supporting the ideas that a) Bin3 associates with components of the telomerase holoenzyme and b) this association is dependent on Pof8.

Fig. 4A: there is no TER1 Northern blot signal in the inputs.

We have provided an additional panel containing semi-quantitative RT-PCR and qRT-PCR (Figure 4A,B).

Fig. 5A and B: statistical analyses should be performed with one-way ANOVA and post hoc test.

We have included a supplemental data table with statistical analyses for total RNA qRT-PCR data (Supplemental Table 10).

Reviewer #2:

In the current manuscript, Porat et al. provide evidence that fission yeast Bin3 interacts with fission yeast telomerase via LARP7 protein Pof8. While I do find their overall findings to be intriguing and exciting, I found multiple issues with the current study that I would like them to address as listed below. Overall, I find evidence that Bin3 interacts with an active telomerase

complex via Pof8 to be convincing. However, I do not believe that authors convincingly established functional significance of Bin3-telomerase interaction, as I found evidence that Bin3 negatively regulate telomerase-dependent telomere elongation from Southern blot analysis (Fig.5) to be preliminary and lack critical controls.

We thank the reviewer for the positive comments on our manuscript and have attempted to address the functional significance of Bin3 in telomerase with additional experiments.

Experimental and data interpretation issues:

1. Supplemental Figure 1C and 1D are lacking control cells to fully evaluate how wt or bin3Δ cells carrying GST-Bin3 plasmid grow in the presence or absence 5FOA or thiamine (B1). It appears that wt cells carrying GST-Bin3 plasmid grow much worse than bin3Δ cells with the plasmid? It would be more useful and easier to compare growth behavior if serial dilution spots for (1) wt+empty plasmid, (2) wt+GST-Bin3 plasmid, and (3) bin3Δ+GST-Bin3 plasmid are plated on the same plate side-by-side on (1) EMM-ura, (2) EMM-ura+B1, or (3) EMM+5-FOA. As it is shown, it is not even clear if panels shown on supplemental figure 1D come from the same plate or pictures taken on the same day for different strains. I am also not certain the rationale of expressing GST-tagged Bin3, rather than non-tagged version of Bin3, as any phenotype associated with overexpression could be attributable to expression of mutant (i.e. GST-tagged) Bin3, rather than simply increasing expression of wt Bin3 protein.

We greatly thank the reviewer for pointing this out. While it is unclear why our original knockout appeared lethal, we are confident that our new knockout is indeed viable (see Figure S5 for validation of the new strain). Since we are now working with a complete knockout, we no longer require plasmids or thiamine to modulate expression.

We have also taken the reviewers' suggestions and moved away from using GST-tagged Bin3 and have repeated experiments using untagged Bin3 in pRep4x.

2. RIP-Seq data presented in supplemental Table for Bin3 seems to be wildly different for 2 replicates for Bin3-PrA IP. In one experiment, Ter1 RNA (SPNCRNA.214.1) and U6 RNA are indeed highly represented, while 2nd experiment found Ter1 or U6 to be barely above background non-tag control. It seems to me that enrichment of Ter1 or U6 would not be counted as significant in replicate 2 if this result is independently analyzed. However, by taking average values of 2 experiments, Figure 1A would give false impression that U6 and Ter1 are reproducibly identified in both replicates of RIP-Seq experiments. Thus, while authors provide convincing follow-up data to substantiate the notion that Bin3 do indeed interacts with U6 and Ter1 RNAs, I would encourage authors to plot data from replicate 1 and replicate 2 of Bin3 RIP separately (at least in supplemental data). It is more honest reflection of actual data, and perhaps authors can speculate why 2nd replicate did not work well, but results from replicate 1 where very high enrichment of U6 and Ter1 were observed in RIP analysis could still be used to rationalize more detailed follow-up experiments described in the manuscript.

We appreciate the thoroughness of the reviewer in pointing this out. While TER1 was enriched in both replicates, there were certainly differences between the replicates, most likely due to RNA degradation during workup and library preparation. We also highly appreciated the comment that the previous RIP-Seq analysis would be sufficient to justify the subsequent follow up work on TER1 and U6. After reflection, however, we were concerned the data set would not be as useful to those in the scientific field that might be interested in Bin3 targets other than TER1. We thus repeated the RIP-Seq in biological triplicate, and the data from the new runs were very consistent both with each other and the interpretations in our original submission. The revised enrichment analysis is presented in Figure 1A and Supplementary Table 1, and below we present, for the benefit of the reviewers, the correlation of reads between the three Bin3 IP samples:

I am also not entirely sure what exactly is plotted in Figure 1A as I don't quite understand labels for X- and Y-axis. What exactly are Log₂ (Bin3 IP +2) and Log₂ (Control IP +2) represent? Is there a significance to "+2"? Perhaps authors can include actual Log₂ values used in making Figure 1A plot in supplemental data table (and leave in formula for generating such values in Excel sheet) so that it is more obvious how data is analyzed.

To conform to literature standards for reporting RIP-Seq data, we now present volcano plots graphing log₂ of fold change (Bin3 IP relative to control IP) against -log of FDR (Figure 1A).

Supplemental Table for RIP-Seq is also difficult to understand and not very accessible. Rather than only listing systematic name, common/functional names of RNA should be given where such names are available. For example, SPNCRNA.214.1 should be made clear that it is Ter1, while SPSNRNA.06.01 should also be listed as "snu6 / small nuclear RNA U6", according to the description given in Pombase. Likewise, snoRNA, tRNA, rRNA, etc. should be clearly identified. There are also large number of "ENSRNA" listed in this table. I am not familiar with such RNA, and I would appreciate some explanation. Are ENSRNAs also included in Figure 1A as well, or only well-defined RNA? The supplemental table also includes some "dubious" RNA (for example SPCC417.15.1) according to Pombase. Are such RNAs also included (or excluded) in Figure 1A? I would also suggest replacing 25_yAS99, 26_yAS99, 31_Bin3, 32_Bin3, etc. with something more descriptive such as no tag control IP, Bin3 IP, etc. in supplemental Table 1.

Having repeated the RIP-Seq we have adjusted the analysis pipeline to a different genome build that more accurately annotates the enriched transcripts (Supplementary Table 1). This revised table also includes the common gene names where known.

3. Lines 172-3: While Bin3-associated U6 snRNA was not immunoprecipitated by the anti-TMG antibody... / I find this statement misleading as U6 signal is clearly visible in TMG IP sample. While it is true that anti-TMG seems to be less efficient in depleting U6 RNA compared to U5 RNA in flowthrough fraction, signal for U6 is only marginally weaker than U5 in TMG IP lane. Thus, authors cannot claim that anti-TMG antibody does not IP U6 RNA. It is either that anti-TMG antibody can cross-react to gamma-monomethyl phosphate cap OR there is a subfraction of U6 carrying in TMG cap, based only on data presented in Fig. 1C-D. If U6 RNA was previously established to be exclusively capped by monomethyl-cap, then it would suggest that anti-body is not as specific to TMG as suggested by authors? Furthermore, I think it would be too strong of a statement to suggest that "TER1 is not a substrate for Bin3-catalyzed modification" (Line 179) unless author can more robustly demonstrate complete absence of monomethyl-cap by utilizing antibody specific to monomethyl-cap or by some other sensitive method to detect monomethyl cap on RNA.

As described in our response to reviewer 1, we have performed the TMG IP under more stringent conditions designed to prevent U6 being immunoprecipitated as a result of an RNA-RNA interaction with other TMG-capped ncRNA. Our new northern blots more accurately reflect previous data indicating a gamma-monomethyl cap on U6 snRNA (Gu, J., Patton, J. R., Shimba, S. & Reddy, R. Localization of modified nucleotides in Schizosaccharomyces pombe spliceosomal small nuclear RNAs: Modified nucleotides are clustered in functionally important regions. *RNA* (1996).).

We agree with the reviewer that our results, in the absence of a more sensitive and specific method to detect the gamma-monomethyl cap, do not conclusively point towards a catalytic-independent role for Bin3 in the context of telomerase RNA, although they do strongly indicate such a role. We have adjusted our wording accordingly. The revised text now reads (lines 162-170), "While Bin3-associated U6 snRNA was not immunoprecipitated by the anti-TMG antibody, in agreement with data demonstrating the presence of a γ -monomethyl phosphate cap on U6 in *S. pombe*³⁵, Bin3-associated TER1 RNA was effectively enriched by anti-TMG immunoprecipitation. These data are consistent with previous work demonstrating a TMG cap on TER1 in *S. pombe*, and also with Bin3 associating with TER1 following spliceosomal cleavage and 5' TMG capping, similar to what has been posited for Pof8¹³. This suggests that Bin3 interacts with the primary cohort of TMG capped TER1 transcripts, and that TER1 is not a substrate for Bin3-catalyzed γ -monomethyl phosphate capping."

4. Supplementary Tables 2 & 3 are incomplete for systematic gene names for fission yeast as they only list cosmid names (which includes multiple genes) and do not indicate individual ORF. (For example, Bin3 should have been indicated as SPBC2A9.10, rather than SPBC2A9.) This omission significantly reduces usefulness of these tables. To make sure readers can clearly identify proteins that are found in mass spec. analysis, unique systematic names should be

included for all genes, even for those that have been named. Furthermore, authors should update protein description for listed genes as some of them seems to be outdated. For example, "uncharacterized RNA binding protein SPBC1861" should have been listed as prp24 and SPBC1861.04c, according to the latest description given in Pombase. (I suppose the version of fission yeast protein database used in mass spec. analysis was somewhat outdated, and thus used old name/descriptions?)

We thank the reviewer for pointing this out. In our revised manuscript we have combined the previous Supplementary tables 2 & 3 into a more comprehensive Supplementary table 2 in which both gene names and uniprot identifiers as well as gene descriptions (see under FASTA headers) are provided for all mass spectrometry listed proteins.

5. Line 208: Authors mention that "the Bin3-Trt1 interaction is lost" upon treatment with benzonase. While I find that Bin3-Trt1 interaction is reduced in benzonase treated sample, residual interaction is clearly visible. Thus, I would suggest that authors modify their statement to reflect only partial disruption of Bin3-Trt1 interaction is found in benzonase treated sample, as "lost" to me imply that the interaction is completely disrupted. Rather than treatment with benzonase, authors might also consider monitoring Bin3-Trt1 interaction in ter1Δ cell background, which might more completely disrupt Bin3-Trt1 interaction if Ter1 RNA is essential for mediating Bin3-Trt1 interaction.

We thank the reviewer for this suggestion and have repeated the Co-IPs both with benzonase and in a *ter1Δ* strain. Both conditions result in a complete loss of the Bin3-Trt1 interaction, which is more readily apparent in our updated western blots (Figures 3 B,C).

6. Lines 229-231: Authors concluded that Bin3-Ter1 interaction is dependent on Pof8 based on loss of Ter1 signal for Bin3 IP in pof8Δ background. However, since previous studies have determined that pof8Δ cells show greatly reduced Ter1 RNA in cells, it is not entirely clear if northern blots shown in Fig. 4A is sensitive enough to detect Bin3-Ter1 interaction. It would also be good to determine what fraction of input RNA is pulled down in Bin3 IP in wt vs pof8Δ cells to control for greatly reduced Ter1 level in pof8Δ cells, but since input Ter1 show no signal, it might not be possible to determine such values by northern blots. Thus, it would be worth monitoring Bin3-Ter1 interaction by utilizing RT-PCR to determine efficiency of Ter1 IP for wt vs. pof8Δ cells.

We have provided semi-quantitative RT-PCR and qRT-PCR data (Figure 4A,B) and the resulting graphs represent the percent of TER1 pulled down relative to an untagged control (Figure 4B).

7. Lines 255-256: Authors suggest that loss of telomerase activity in Bin3 pull down in pof8Δ cells support the notion that Pof8 is required for Bin3-telomerase interaction. I don't think this is a valid argument since Peter Baumann's group has shown that in vitro telomerase activity is barely detectable in pof8Δ cells even when Trt1 IP samples were used in assay. Since pof8Δ cells

show greatly reduced active telomerase, it would have been nearly impossible to detect telomerase activity in pof8Δ cells even if Bin3 is still robustly interacting with Ter1.

We thank the reviewer for pointing this out and have reworded this section accordingly. The text now reads (lines 251-254): “Consistent with previous results^{13,16}, we also observed a loss of activity for Bin3 immunoprecipitated from a pof8Δ strain, which can largely be attributed to the loss of functional, correctly assembled telomerase occurring in the absence of Pof8.”

8. Lines 265-268 & supplementary Fig. 4: Authors state that overexpression of Bin3 does not rescue telomere shortening and telomere fusions observed in pof8Δ cells, but only show PCR assays for detecting telomere fusion and no data on how Bin3 overexpression affects telomere length by Southern blot. I think authors should monitor telomere length before claiming that Bin3 over-expression does not rescue telomere shortening in pof8Δ cells.

We have provided the corresponding southern blots in Supplementary figure 4B.

On a minor note, plasmid name indicated in this figure does not match other figures. Does this figure utilize non-GST-tagged plasmid, or same pREP4x-GST-Bin3 plasmid as other figures? Strain list (supplementary Table 5) for this figure does not actually include plasmid, so I am not sure exactly what is shown in supplementary Fig. 4

We have taken care to ensure clear figure labels in the updated figures (i.e. empty plasmid= pRep4x).

9. Based on data shown in Fig. 5, authors argue that Bin3-telomerase interaction negatively regulates telomerase-dependent telomere elongation by negatively regulating TER1 accumulation in cells. However, I am not entirely convinced if claimed slight elongation of telomeres in cells expressing reduced level of GST-Bin3 is significant based on Southern blot presented in Fig. 5C. There are several issues/suggestions to improve robustness of Southern blots as listed below.

We have repeated the experiments in figure 5 with our new knockout and demonstrate that similar to Pof8, Bin3 has a role in promoting telomere elongation. These data are even more convincing using the Apal digest suggested by this reviewer.

(a) Since average telomere length maintained by cells are very sensitive to growth condition and even wt cells show changes in average telomere length depending on media used to grow cells, temperature, etc. It is critical that authors include additional controls to ensure that growing cells in the presence (+B1) or absence (-B1) of thiamine did not cause telomere length change in control cells. Authors should minimally monitor telomere length for cells grown in same number of generations in same media with or without added thiamine (B1) for (1) wt cells carrying empty pREP41-GST plasmid, (2) wt cells carrying pREP41x-GST-Bin3, (3) bin3Δ cells carrying pREP41x-GST-Bin3. For a given genotype and growth condition, multiple independently

derived clones should be utilized as well since plasmid copy number can vary greatly among independent transformants in fission yeast.

In our updated southern blots, we are working with endogenous Bin3 expression (or lack thereof), so differences in plasmid copy number are not an issue. Such a change also dismissed the need for (+) and (-) thiamine controls.

(b) Quality of gDNA used in Fig. 5C seems questionable as wt cell lane contain unexplained 2nd peak of telomere signal much smaller than 1 kb, which is not usually seen in Southern blot for EcoRI digested gDNA (Wt cells should show peak of telomere signal ~1 kb for telomeres.) I suspect this might be caused by degradation of gDNA. For unknown reason, when fission yeast cells are grown in minimum media, we have also seen appearance of these smaller band if culture is over-saturated. Thus, it would be worth repeating gDNA prep by ensuring that cells are still in log-phase. Another issue with gDNA prepared from cells grown in minimum media is that EcoRI digestion would frequently fail if starting concentration of gDNA is too low. Such incomplete digestion would often produce appearance of much larger product on Southern blot. For bin3Δ+4x GST-Bin3+B1 sample, there appears to be 2 distinct peaks with much bigger product is observed, and I suspect this is due to partial digestion of gDNA. To avoid partial digestion, authors should prepare gDNA samples from larger culture (and ensuring exponential cell growth when samples are collected).

We thank the reviewer for these suggestions. We have repeated southern blots with gDNA grown from larger cultures we ensured were still in log phase. While we do not observe any larger species (incomplete digestion), we were unable to completely eliminate the smaller species in our EcoRI digest. However, we feel the expected telomere size is predominant and clearly demonstrates a shortening of telomeres in the Bin3 deletion strain, an observation that is even more apparent in the Apal digested genomic DNA (see below), for which there are no unexpectedly smaller species.

(c) As I mentioned earlier when discussing results shown in supplementary Fig. 1, I am not entirely sure why authors are using GST-tagged Bin3 expression plasmid if they want to examine effect of reduction in Bin3 expression. I think that use of pREP41 plasmid without tag would be preferred to make sure that GST-Bin3 is not causing unexpected dominant phenotype. I also wonder if authors may be able to obtain more clear change in telomere length if they were to utilize pREP81 series plasmid, which should express even less Bin3. Did authors found that pREP81-Bin3 would not be allow bin3Δ cells to survive?

Please see our above comment regarding GST-Bin3. Since we have successfully constructed a knockout, we also did not need to further modulate expression with different promotor strengths (but nevertheless still thank the reviewer for these helpful suggestions, all of which have improved the quality of our southern blots).

(d) I do see that perhaps very slight shift in peak of telomere signal that are found between 1 kb and 1.5 kb ladder. However, resolution of EcoRI digested Southern blot is not sufficient to see

such small change in telomere length. To see slight changes in telomere length better, Apal digest should be used in Southern blot since average telomere length in Apal-digested Southern blot is much smaller (~300bp) and thus able to better resolve small changes in telomere length. Quantification of telomere peak would also be helpful.

Again, we thank the reviewer for these insightful tips! We now present southern blots from EcoRI- and Apal-digested gDNA and indeed see a more dramatic difference in telomere length between strains, indicative of Bin3 having a role in promoting telomerase activity (Figure 5B). We feel this change in telomere length is more readily apparent relative to our original submission.

(e) Labels for Fig.5 is confusing. For 5A and 5B, bin3Δ cells are presumably carrying pREP4x GST-Bin3 plasmid, but this is not indicated. For Fig.5C, I assume that wt and pof8Δ cells are carrying empty pREP4x-GST plasmid as both strains are indicated with “4x”, but strain table (supplementary Table 5) does not indicate any plasmid for wt and pof8Δ cells. Do wt and pof8Δ cells carry plasmid or not?

Our updated figure is working with endogenous Bin3 and Pof8 levels in rich media and as such, these new strains do not carry plasmids.

10. Lines 339-340: As I discussed on point 6 above, it is not entirely clear if “interaction between Bin3 and TER1 is dependent on Pof8” since pof8Δ cells carry greatly reduced TER1 RNA and it is not clear Northern blot can detect reduced TER1 RNA in Bin3 pull down.

See our above comment to reviewer 1, but we have now provided qRT-PCR data to support our northern blots (Figure 4B).

11. Lines 370-373: Authors seem to suggest that inability to limit telomerase activity in be absence of Bin3 might be the cause to unviability for bin3Δ cells since deletion of tpz1 or stn1 are also “essential” proteins. I don’t think this is a valid argument and it should be removed. First of all, “essential” function for Tpz1 and Stn1 lies with telomere capping function, and second, viable tpz1Δ and stn1Δ cells can be obtained since survivor cells that carry circular chromosomes can be readily obtained in fission yeast. It seem to be that actual cause of bin3Δ inviability probably lies with its function in regulation of U6 RNA as many of splicing factors are essential in fission yeast.

We thank the reviewer for highlighting this and have removed speculation as to the essentiality of telomerase-associated proteins.

12. Lines 384-385: Authors mention the study that found reduced telomere length for LARP7 mutant patient to suggest that role of LARP7 in telomerase regulation might be conserved in

humans as well. However, the cited study found that LARP7 patient show alteration in TERT splicing, which is likely the cause of telomere phenotype. To date, huma LARP7 has not been identified as an integral part of telomerase holoenzyme complex, and recent studies have found functional roles for LARP7 in U6 RNA regulation, it seems rather unlikely that human LARP7 would play conserved role in telomerase holoenzyme complex.

We have removed this section from the manuscript and instead refocused our speculation on the conservation of LARP7 around its role in U6.

Minor editorial issues:

1. For introduction section, authors could improve in citation of previous research. Here are some examples. For line30: 1st sentence could use some references. For line35: also cite Nakamura et al. Science 19997 for S. pombe Trt1. For line39: authors should cite primary papers (or reviews) which established that yeast and metazolan telomerase RNA is transcribed by RNA Pol II. For line68: authors should include Juli Feigon papers where her group identified and described xRRM motif.

2. Figure legend for Fig 1D should also mention U6 and U5.

3. Figure legend for Fig. 2B should also mention U6 data.

We have updated citations and figure legends.

Reviewer #3:

In this manuscript, the Bayfield group investigates the biochemical composition of the S. pombe telomerase holoenzyme. They report on the unexpected observation that the capping enzyme Bin3 is a stable component of this Sm/Lsm type RNP. The authors further provide evidence that Bin3 associates apart from the U6snRNA also with the telomerase holoenzyme via an interaction with the protein Pof8. Importantly, this factor belongs to the Larp7 family, which has been linked to 7SK RNP and more recently also to U6 modification. The data clearly show that Bin3 does not generate the cap of TER1 but rather affects the composition, assembly, and regulation of the telomerase holoenzyme in yeast.

This is a potentially interesting study providing valuable new insight into telomerase biology and its evolution.

We thank the reviewer for their positive assessment of our manuscript.

The following aspects should be addressed to further support the conclusions drawn by the authors:

1. It is not clear to me whether the particle schematically shown in Fig. 4D exists with the indicated set of proteins or whether this is actually a mixture of particles with different protein compositions. If technically possible, the authors should add (biochemical) experiments that further illustrates the composition of the newly defined particle.

We constructed the schematic of the telomerase holoenzyme based on the previous model proposed by the Nakamura group (Mennie, A. K., Moser, B. A. & Nakamura, T. M. LARP7-like protein Pof8 regulates telomerase assembly and poly(A)+TERRA expression in fission yeast. *Nat. Commun.* (2018) doi:10.1038/s41467-018-02874-0.) as well as the list of proteins recovered in our co-immunoprecipitation mass spectrometry (Figure 3A, supplemental tables 2 and 3). We agree that it would be interesting to examine potential heterogeneity among telomerase holoenzymes, either *in vivo* or through *in vitro* reconstitution, although we are not aware a yeast telomerase has been successfully reconstituted. As an additional experiment to complement our mass spectrometry, Co-IP western, and telomerase assay data, we have performed glycerol gradient sedimentation to demonstrate that Bin3, Pof8, and TER1 co-migrate in a Pof8 dependent manner, consistent with complex formation.

2. In general, I found the quality of several Northern and Western blots presented in the manuscript of low quality, which makes the interpretation of the data difficult. Sometimes, the bands are not even visible (see for example 1 B, 2D). The respective blots should be replaced. Input lanes often have a weaker or no signals than the IP lanes (see 1C, 4A. I assume that only a small fraction of the inputs is loaded, which should be mentioned). Fig. 2D is not mentioned in the text.

We have updated the text with a reference to figure 2D.

As mentioned in our response to reviewers 1 and 2, TER1 is a very lowly expressed RNA and as such, northern blotting is often not sensitive enough to detect TER1 in input RNA. We have added semi-quantitative PCR gels and qRT-PCR to the existing northern blots. 1% inputs were run on northern blots, which we have clarified in the updated materials and methods section.

We have also repeated our western blots with optimized antibody concentration and chemiluminescent detection means to improve their quality.

3. The enzymatic activity assayed in Fig. 4C is not properly controlled as it is not clear whether comparable amounts of material (Trt1, Pof8, Bin3) has been included in this assay. The quality of this experiment can also be improved as this is a well-established assay. The lower panel of this figure is not labelled.

Since we are using different antibodies to immunoprecipitated different components (i.e. Bin3 and Pof8), there is no way to directly compare activity associated with these varying IPs. That said, it is common practice to normalize telomerase extension signal to a loading control (which

we have also labeled in the bottom panel). In the revised manuscript, we directly compare only telomerase activity relative to the same IP (Bin3 in Figure 4D, Pof8 in Figure 5D). We have also normalized telomerase extension products (Figure 5D) to TER1 levels and our updated quantifications now represent relative telomerase activity (normalized to wild type TER1 levels).

We thank all three reviewers again for their very helpful comments and hope that we have satisfactorily addressed their concerns.

REVIEWERS' COMMENTS

Reviewer #1 (Remarks to the Author):

The authors have responded to my comments in an adequate and detailed manner. The manuscript now contains a range of additional experiments and has improved substantially.

Minor comments:

Fig. 4B legend: the authors probably mean „Relative TER1 and U6 IP was calculated by comparing percent immunoprecipitation of TER1 or U6 to immunoprecipitation from an untagged strain“

Fig. 4B, 5A and 5C legend: it should be „* at $p < 0.05$ “

Reviewer #2 (Remarks to the Author):

Overall, I found the revised manuscript from Porat et al. to be greatly improved, and generally support the main conclusion that SPBC2A9.10 is stably associated with Pof8 and contributes to telomerase assembly in fission yeast. I am also generally satisfied with responses to reviewers by authors. However, I do still have some remaining issues that I would like authors to address as listed below.

1. I would urge authors to come to agreement with Dr. Peter Baumann's group to use same name for SPBC2A9.10 (Bin3 or Bmc1) to avoid confusion in the future.
2. It would be helpful for readers if authors clearly indicate names of supplemental tables and descriptions within Excel or PDF files. Supplemental Table 2 is also extremely difficult to get usable information as it only lists protein IDs but not gene number or known gene name. For example, telomerase catalytic subunit Trt1 is only mentioned as "sp|O13339|TERT_SCHPO" and never mentioned by official gene name (Trt1) or systematic name (SPBC29A3.14c). Thus, I hope authors can revise supp. Table 2 to include common name (if available) and systematic names, not just protein ID.
3. Lines 192-194 mentions that uncharacterized protein (SPCC18B5.09c) retained interaction with Bin3 in the presence of benzoase, but there is no data shown at all for this. While SPCC18B5.09c has been identified by Baumann's groups as integral component of telomerase in accompanying submission, but this has not yet been officially published, so I am bit puzzled why authors are mentioning this here as "componet of telomerase complex." The only place "SPCC18B5.09c" does show up is within supp. Table 2 "Fasta header" section for protein ID of "sp|Q9USL2|YJK9_SCHPO" (Third top hit), but it very difficult to see this information. (Without any actual data, I would suggest to simply remove lines 192-194.)
4. Authors should try to define abbreviations more clearly in main text and methods, etc. For example, RIP-Seq, CPM, FDR, FC, F, TMM, etc. are just mentioned/used without any explanation on what they are

stand for, and it is not always clear what these abbreviations...

5. While I understand TER1 is difficult to detect by Northern blot, but I don't quite understand why Bin3 ProA and Bin3 ProA pof8Δ lanes appear to show similar signal for TER1 Input. Based on previous publications, TER1 level should be substantially reduced in pof8Δ cells. Thus, I wonder if RT-PCR data is not in linear range of PCR and does not really serve as "Input" control.

6. Glycerol gradient data shown in supp. Fig.3 is quite low quality and I don't quite see what authors describe in the main text lines 231-241. While the strongest signal for TER1 for "semi-quantitative TER1 RT-PCR" is seen for right most lane, but there seems to be very little to no signal for Bin3 and Pof8. Thus, I don't quite see what authors refer to as "co-sedimentation" of Bin3, Pof8 and TER1... It is also not clear how reproducible this data is as it seems that RT-PCR might be inhibited in some lanes (for example, 2nd lane from right for U2). In its current form, I think it is perhaps better to just remove this data.

8. Lines 248-254 could be improved by first mentioning Fig.4D when discussing in vitro telomerase assay using Bin3 IP. As it is written, I looked at Fig.5D and got confused that data shown is not for Bin3 IP but for Pof8 IP telomerase activity assay.

9. Figure legends for Fig.4B, 5A, 5C, and supp. Fig.5B-C mention "*" at $p > 0.05$ ", but I think it should be " $p < 0.05$ " if it is marked as statistically significant. In addition, "n.s." should be defined as not significant at least in the first time it is used.

10. For Fig.5A, supp.Fig.5B,D, "normalized" wt lane is not equal to "1". Why are these numbers not 1 but some other values? Are they meant to be different from 1?

8. I am not entirely sure what authors mean by "prediction 67" of Phyre2 structure prediction (supp. Fig.1C) and why it is important to mention this? Does prediction 67 supposed to mean something? On the other hand, authors should cite and acknowledge Phyre2 as suggested at Phyre2 website.

Reviewer #3 (Remarks to the Author):

In this extensive revision, the authors have addressed all points raised by the three referees in a satisfactory manner. They have included new experiments that strengthen their major conclusions and also improved the overall quality of several experiments. I therefore support publication of this interesting study.

Reviewer #1 (Remarks to the Author):

The authors have responded to my comments in an adequate and detailed manner. The manuscript now contains a range of additional experiments and has improved substantially.

We thank the reviewer for their support of our manuscript.

Minor comments:

Fig. 4B legend: the authors probably mean „Relative TER1 and U6 IP was calculated by comparing percent immunoprecipitation of TER1 or U6 to immunoprecipitation from an untagged strain“

Fig. 4B, 5A and 5C legend: it should be „* at $p < 0.05$ “

We thank the reviewer for pointing these out. Both typos have been fixed.

Reviewer #2 (Remarks to the Author):

Overall, I found the revised manuscript from Porat et al. to be greatly improved, and generally support the main conclusion that SPBC2A9.10 is stably associated with Pof8 and contributes to telomerase assembly in fission yeast. I am also generally satisfied with responses to reviewers by authors. However, I do still have some remaining issues that I would like authors to address as listed below.

We are glad the reviewer responded favorably to our revisions.

1. I would urge authors to come to agreement with Dr. Peter Baumann's group to use same name for SPBC2A9.10 (Bin3 or Bmc1) to avoid confusion in the future.

We have clarified this in our title, abstract, and manuscript text by referring to SPBC2A9.10 as Bcm1 to remain consistent both with Dr. Baumann's manuscript (which is also referenced in our manuscript to highlight the complementary narratives that have emerged from both of our groups).

2. It would be helpful for readers if authors clearly indicate names of supplemental tables and descriptions within Excel or PDF files. Supplemental Table 2 is also extremely difficult to get usable information as it only lists protein IDs but not gene number or known gene name. For example, telomerase catalytic subunit Trt1 is only mentioned as "sp|O13339|TERT_SCHPO" and never mentioned by official gene name (Trt1) or systematic name (SPBC29A3.14c). Thus, I hope authors can revise supp. Table 2 to include common name (if available) and systematic names, not just protein ID.

The gene names are provided in the Fasta header column (i.e. GN= Trt1). The freely accessible mass spectrometry analysis software we have used does not output gene names, only protein IDs. Since Supp Table 2 only lists the top 50 hits, we have manually added the gene names in a new column.

3. Lines 192-194 mentions that uncharacterized protein (SPCC18B5.09c) retained interaction with Bin3 in the presence of benzoase, but there is no data shown at all for this. While SPCC18B5.09c has been identified by Baumann's groups as integral component of telomerase in accompanying submission, but this has not yet been officially published, so I am bit puzzled why authors are mentioning this here as "componet of telomerase complex." The only place "SPCC18B5.09c" does show up is within supp. Table 2 "Fasta header" section for protein ID of "sp|Q9USL2|YJK9_SCHPO" (Third top hit), but it very difficult to see this information. (Without any actual data, I would suggest to simply remove lines 192-194.)

Again, we thank the reviewer for mentioning this. The benzonase data was included in a version of supplemental table 2 included in our original submission, which had since been replaced with the current supplemental table 2. We have updated the table with this information.

Since our original submission we have been in contact with Peter Baumann's lab and learned of his lab's identification of the telomerase protein Thc1, the uncharacterized protein we identified in our mass spectrometry data. We have updated this with a reference to his lab's pre-print. The revised manuscript now reads, (lines 193-197) "With the exception of Pof8, certain members of the Lsm2-8 complex, and an uncharacterized protein (SPCC18B5.09c, recently identified as the telomerase component Thc1⁴¹), interactions between Bin3 and other components of the telomerase holoenzyme were lost, suggestive of an interaction mediated by TER1 (Table S2)."

4. Authors should try to define abbreviations more clearly in main text and methods, etc. For example, RIP-Seq, CPM, FDR, FC, F, TMM, etc. are just mentioned/used without any explanation on what they are stand for, and it is not always clear what these abbreviations...

We thank the reviewer for pointing this out and have now defined abbreviations when they are first mentioned.

5. While I understand TER1 is difficult to detect by Northern blot, but I don't quite understand why Bin3 ProA and Bin3 PrA pof8Δ lanes appear to show similar signal for TER1 Input. Based on previous publicastions, TER1 level should be substantially reduced in pof8Δ cells. Thus, I wonder if RT-PCR data is not in linear range of PCR and does not really serves as "Input" control.

The reviewer is correct that RT-PCR may not be in the linear range for Figure 4A (even if it does appear that TER1 signal is lesser in the dPof8 input lane, relative to wt), but with the accompanying qRT-PCR data in Figures 4B and 5A that shows the reported decrease in TER1 levels in the pof8Δ strain, we were able to quantitatively determine that even with decreased

TER1 levels, Bin3 nevertheless loses its interaction with TER1 in the absence of Pof8.

6. Glycerol gradient data shown in supp. Fig.3 is quite low quality and I don't quite see what authors describe in the main text lines 231-241. While the strongest signal for TER1 for "semi-quantitative TER1 RT-PCR is seem for right most lane, but there seems to be very little to no signal for Bin3 and Pof8. Thus, I don't quite see what authors refer to as "co-sedimentation" of Bin3, Pof8 and TER1... It is also not clear how reproducible this data is as it seems that RT-PCR might be inhibited in some lanes (for example, 2nd lane from right for U2). In its current form, I think it is perhaps better to just remove this data.

We appreciate the reviewer's concerns about this figure. We performed gradient centrifugation to address comments brought up by reviewers 1 and 3 suggesting additional biochemical means of validating the presence of Bin3 and Pof8 in the telomerase RNP. We observe Bin3, Pof8, and TER1 in lanes 6 and 7, suggesting co-sedimentation of the 3 in a complex, while Bin3 is absent from lanes 6 and 7 in the *pof8Δ* strain. This is suggestive of complex formation mediated by Pof8, as without Pof8, Bin3 and TER1 show impaired co-sedimentation. As we have also shown that Bin3 and Pof8 are present in a U6-containing complex, we anticipate that Bin3 and Pof8 migrating in lighter fractions towards the left of the blot may represent Bin3 and Pof8 on U6.

We also acknowledge the reviewer's concern about reproducibility of the figure. We have performed this experiment several times and chosen a representative image to include in the manuscript. Below you can find another trial (this gradient spread out over more lanes but otherwise identically performed to the figure in the current manuscript). We observe the same pattern with regards to co-sedimentation: Bin3, Pof8, and TER1 are present in lanes 10-12, while Bin3 is missing in lanes 11 and 12 in the absence of Pof8 and instead migrates in lighter fractions of the gradient.

We have updated the manuscript with this alternate trial, which, due to fractions being more spread out, may clarify our interpretations.

8. Lines 248-254 could be improved by first mentioning Fig.4D when discussing *in vitro* telomerase assay using Bin3 IP. As it is written, I looked at Fig.5D and got confused that data shown is not for Bin3 IP but for Pof8 IP telomerase activity assay.

We thank the reviewer for helping clarify our manuscript and have updated the text accordingly. The revised manuscript now reads (lines 246-252), "We performed a previously described *in vitro* telomerase assay that relies on the presence of the TER1- and Trt1-containing telomerase holoenzyme to extend an oligonucleotide resembling telomeric DNA¹³ (Figure 4D). Bin3 immunoprecipitates extended the oligonucleotide in a similar manner previously demonstrated for Pof8 (¹³ and see Figure 5D), as well as showed the same loss of activity upon RNase A treatment, supporting the idea that Bin3, much like Pof8, is a component of the active telomerase holoenzyme (Figure 4D)."

9. Figure legends for Fig.4B, 5A, 5C, and supp. Fig.5B-C mention "* at p>0.05", but I think it should be "p<0.05" if it is marked as statistically significant. In addition, "n.s." should be defined as not significant at least in the first time it is used.

We thank the reviewer for pointing this out and have corrected it.

10. For Fig.5A, supp.Fig.5B,D, "normalized" wt lane is not equal to "1". Why are these number not 1 but some other values? Are they mean to be different from 1?

For our qPCR calculations, we averaged the Ct values of 3 wild type (wt) biological replicates, each with 3 technical replicates, and compared each replicate to the average using the $\Delta\Delta C_t$ method. The mean of these $\Delta\Delta C_t$ values is presented on the graphs in figures 5A, S5B, and S5D, along with the standard error of the mean.

We have clarified this in our methods section, which now reads, (lines 495-498) "TER1 and U6 levels were normalized to act1 mRNA levels and the average wild type Ct value, and subsequently subject to paired two-tailed student's *t*-tests and, where applicable, one-way ANOVA followed by a Tukey posthoc test with a set to 0.05 (Table S10)."

8. I am not entirely sure what authors mean by "prediction 67" of Phyre2 structure prediction (supp. Fig.1C) and why it is important to mention this? Does prediction 67 supposed to mean something? On the other hand, authors should cite and acknowledge Phyre2 as suggested at Phyre2 website.

The citation is meant to refer to the Phyre2 website, so we have re-ordered the sentence to reflect that. The revised manuscript now reads (lines 904-906), "Phyre2⁶⁸ structure prediction of *S. pombe* Bin3 bound to a 5' terminal guanosine (orange) and SAH (green). Predicted SAM-binding residues are in red and nucleotide-binding residues are in blue." Reference 68 refers to the Nature Protocols article introducing Phyre2 (Kelley et al., 2015).

Reviewer #3 (Remarks to the Author):

In this extensive revision, the authors have addressed all points raised by the three referees in a satisfactory manner. They have included new experiments that strengthen their major conclusions and also improved the overall quality of several experiments. I therefore support publication of this interesting study.

We thank the reviewer for their comments and are pleased they find the study interesting.